# Inverse problems with experiment-guided AlphaFold

**Advaith Maddipatla** [* 1 2 3]  **Nadav Bojan Sellam** [* 1 3]  **Meital Bojan** [* 1 3]
**Sanketh Vedula** [1 3 †]  **Paul Schanda** [3 †]  **Ailie Marx** [4 5 †]  **Alex M. Bronstein** [1 3 †]

## Abstract

Proteins exist as a dynamic ensemble of multiple conformations, and these motions are often crucial for their functions. However, current structure prediction methods predominantly yield a single conformation, overlooking the conformational heterogeneity revealed by diverse experimental modalities. Here, we present a framework for building experiment-grounded protein structure generative models that infer conformational ensembles consistent with measured experimental data. The key idea is to treat state-of-the-art protein structure predictors (e.g., AlphaFold3) as sequence-conditioned structural priors, and cast ensemble modeling as posterior inference of protein structures given experimental measurements. Through extensive real-data experiments, we demonstrate the generality of our method to incorporate a variety of experimental measurements. In particular, our framework uncovers previously unmodeled conformational heterogeneity from crystallographic densities, and generates high-accuracy NMR ensembles orders of magnitude faster than the status quo. Notably, we demonstrate that our ensembles outperform AlphaFold3 (Abramson et al., 2024) and sometimes better fit experimental data than publicly deposited structures to the Protein Data Bank (PDB, Burley et al. (2017)). We believe that this approach will unlock building predictive models that fully embrace experimentally observed conformational diversity.

---

[*]Equally contributing first author. [†]Equally contributing senior author. Correspondence contact: sai.maddipatla@cs.ox.ac.uk, n.sellam@campus.technion.ac.il, meital.bojan@campus.technion.ac.il [1]Technion–Israel Institute of Technology, Israel. [2]University of Oxford, UK. [3]Institute of Science and Technology, Austria. [4]Tel Hai Academic College, Israel. [5]MIGAL - Galilee Research Institute, Israel.

*Proceedings of the 42$^{nd}$ International Conference on Machine Learning*, Vancouver, Canada. PMLR 267, 2025. Copyright 2025 by the author(s).

## 1. Introduction

Proteins are inherently dynamic entities, sampling a continuum of conformational states to fulfill their biological roles. Experimental techniques such as X-ray crystallography, nuclear magnetic resonance (NMR) spectroscopy, and cryo-electron microscopy (cryo-EM) inherently report on ensemble-averaged data rather than singular static snapshots. In X-ray crystallography, the resolved electron density map represents a spatial and temporal average over all molecules in the crystal lattice, with regions of flexibility manifesting as diffuse or poorly resolved density. NMR spectroscopy measures the interaction between nuclear spins (e.g., magnetization transfer due to nuclear Overhauser effect, NOE) and spins and electrons (e.g., chemical shifts) arising from dynamic conformational ensembles in solution, with these experimental restraints used computationally to resolve compatible structural states. Cryo-EM similarly resolves multiple conformational states, as individual particles frozen in vitreous ice adopt distinct orientations and conformations, which are computationally classified into discrete or continuous flexibility ranges.

On the computational front, *ab initio* protein structure determination based on modeling the molecule's free energy and its subsequent minimization (e.g., Rosetta and many of its variants (Baek et al., 2021; 2024)) have been only partially successful and computationally expensive. A giant leap in protein structure prediction resulted from the fundamental discovery of the coevolution of contacting residues (Göbel et al., 1994; Hopf et al., 2014), underlying deep learning-based models such as AlphaFold (Jumper et al., 2021; Abramson et al., 2024), which had a groundbreaking impact on structural biology and was awarded the recent 2024 Nobel Prize in Chemistry.

Protein structure predictors are trained exclusively on X-ray crystallographic models, which are themselves *fitted* to electron density maps averaged over trillions of molecule instances. While it has been recognised several decades ago that the conformations of proteins in crystals are heterogeneous (Smith et al., 1986; Furnham et al., 2006), early crystallographic refinements prioritized single-conformer models. Advances in resolution, the more widespread application of room-temperature crystallographic experiments

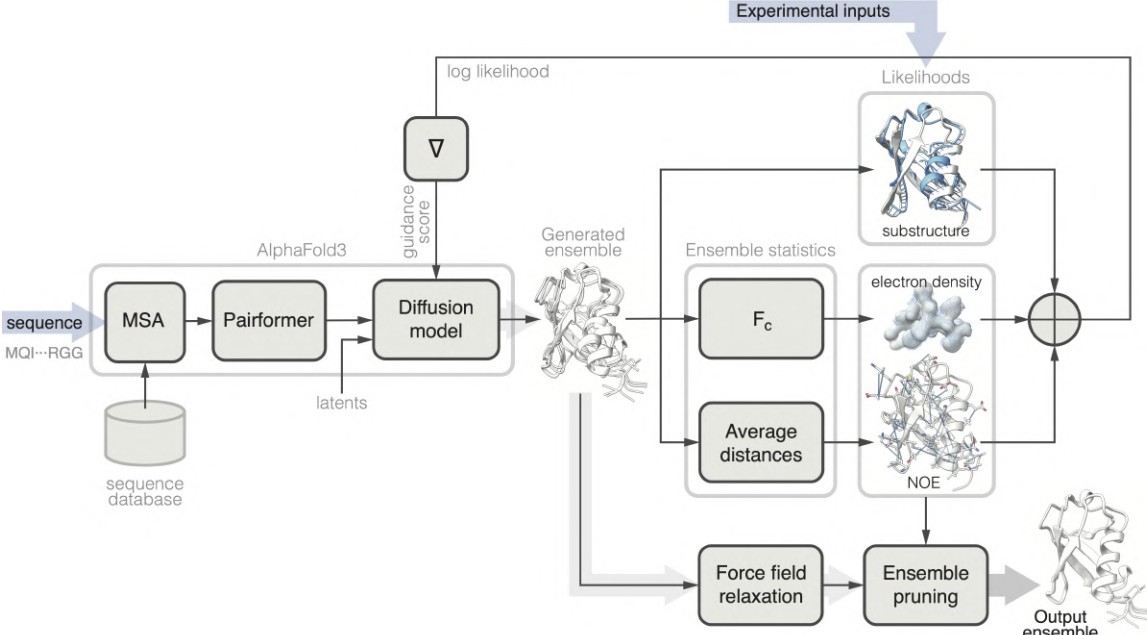

*Figure 1.* **Schematic depiction of the proposed method.** AlphaFold3 allows the sampling of protein structures given an amino acid sequence. To further condition the model by experimental observations, at each time step of the AlphaFold3 diffusion model, an ensemble of structures is generated. Likelihoods of experimental observations are calculated given each individual ensemble member (e.g., to enforce a substructure) and on ensemble averages (e.g., calculated electron density $F_c$ and average inter-atomic distances). The gradient of the combined log-likelihood terms is used as the guidance score. At the final diffusion step, the generated ensemble is refined by force field relaxation and pruned by an orthogonal matching pursuit-like procedure to improve the likelihood terms.

(as opposed to those performed at 100 K), and progress in refinement tools now permit explicit modeling of alternative conformations ("altlocs") within overlapping density regions (Furnham et al., 2006; van den Bedem & Fraser, 2015; Wankowicz et al., 2024). Recent studies analyzing the PDB reveal that such multi-conformer annotations are widespread, reflecting inherent structural variability captured in crystallography (Rosenberg et al., 2024a). In NMR spectroscopy, the experimental observables, such as inter-atomic distances or bond-vector orientations reflect the time- and ensemble average, and NMR structures are always reported as bundles of conformations. However, AlphaFold's training objective – to predict a single "most probable" structure – biases its output toward static snapshots, effectively marginalizing conformational heterogeneity encoded in its training data.

Over the past year, multiple sequence-conditioned protein structure generative models like AlphaFlow (Jing et al., 2024), and the recent AlphaFold3 (Abramson et al., 2024) have been proposed to move beyond the one-sequence–one-structure paradigm. However, since these approaches remain *trained on unimodally-modeled PDB entries* derived predominantly from crystallographic data, the generated ensembles fail to capture the full heterogeneity implied by experimental measurements, thus limiting their practical utility (Rosenberg et al., 2024b). This emphasizes the need

for new models that can explicitly model protein ensembles that are faithful to experimental measurements. Developing such models is the focus of the present work.

## 2. Contributions

In this work, we introduce *experiment-guided AlphaFold3*, a computational framework that integrates experimental data with deep learning priors to generate structural ensembles consistent with experimental observables. Our key insight is that AlphaFold3 can be viewed as a strong sequence-conditioned protein structure prior that may be further leveraged to solve inverse problems in the space of protein structures. By solving these inverse problems under the prior imposed by AlphaFold3, we bridge the gap between data-driven predictions and experimental evidence, yielding ensembles that are both physically plausible and experimentally consistent.

**Experiment-guided AlphaFold3.** Our primary technical contribution is a three-stage ensemble-fitting pipeline (Figure 1). First, we present Guided AlphaFold3, where we adapt the diffusion-based structure module of AlphaFold3 to incorporate experimental measurements during sampling. To properly handle ensemble measurements, we introduce a *non-i.i.d. sampling scheme that jointly samples the ensemble*, directing conformational exploration toward regions

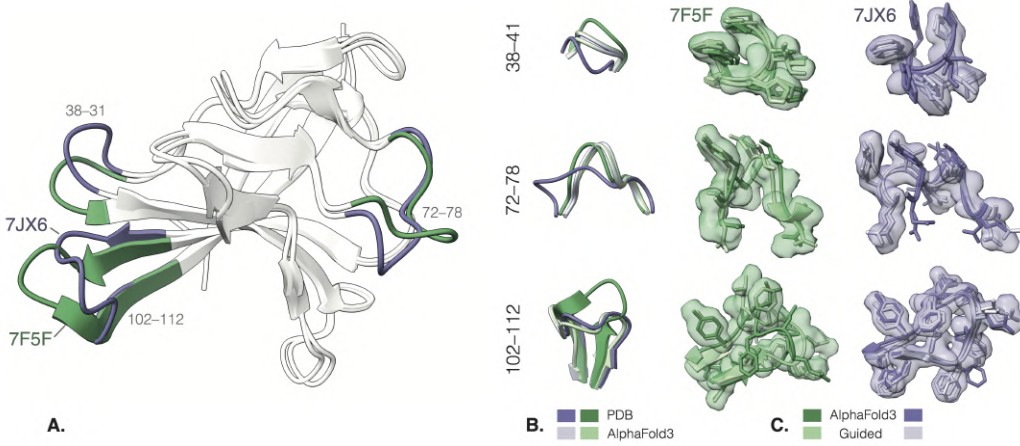

*Figure 2.* Two crystallographic observations of the SARS-CoV-2 ORF8 protein at 1.62Å resolution (PDB: `7JX6` and `7F5F` color coded as purple and green, respectively). The two structures exhibit major differences at three sites despite sharing a nearly identical main acid sequence deferring by a single mutation `L67S` (A). AlphaFold3 prediction appears practically identical for the two sequences mispredicting the structure of the turn at site `38-41`, correctly predicting the structure of `7F5F` and mispredicting that of `7JX6` at site `72-78`, and correctly predicting the structure of `7JX6` mispredicting that of `7F5F` at site `102-112` (B). Electron density-guided AlphaFold3 corrects these predictions by producing ensembles fitting well into the observed electron densities $F_o$ of both structures depicted as the 0.3 $[e^-/Å^3]$-isosurfaces (C).

compatible with the experimental constraints. We show that this approach effectively captures multi-modal ensemble measurements, where standard i.i.d. sampling methods fail (Figure A2). To our knowledge, this represents the first application of guided sampling within AlphaFold3 for experimental structural resolution. Second, we address artifacts introduced during guided sampling by using AlphaFold2's computationally efficient force-field relaxation step, effectively projecting candidate structures onto physically realistic conformations. Finally, we develop a matching-pursuit ensemble selection algorithm to iteratively refine the ensemble by maximizing agreement with experimental data while preserving structural diversity. We validate our framework through case studies on two foundational challenges in structural biology: (1) X-ray crystallographic structure modeling, where we recover conformational heterogeneity obscured in static electron density maps, and (2) NMR structure determination, where we resolve ensembles that obey NOE-derived distance restraints.

**Improved crystal density modeling.** X-ray crystallography is one of the most accurate techniques for protein structure determination. A typical pipeline involves the crystallization of protein samples and the subsequent fitting of atomic structures to electron density maps generated from X-ray diffraction patterns. However, this procedure is expensive, time-consuming, and often requires manual intervention by crystallographers (Doudna, 2000). As a result, several structures deposited in the PDB exhibit human-induced biases that can degrade structural accuracy. Another limitation of crystallographic pipelines is the misleading notion of "single crystal and single structure". However, the PDB exhibits

multimodality in the density that cannot be fully captured by models like AlphaFold3 that predict single structures. This limitation, recognized early on in protein crystallography (Smith et al., 1986), is particularly evident in *altloc regions* (Rosenberg et al., 2024b), where multiple conformations coexist in the same lattice (van den Bedem & Fraser, 2015; Furnham et al., 2006). This inadequacy presents a compelling case for protein generative models to improve crystallographic structural modeling.

Hence, we introduce *Density-guided AlphaFold3*, which guides AlphaFold3-generated structural ensembles to be faithful to experimental electron density maps. Density-guided AlphaFold3 renders structures that are consistently more faithful to the observed electron density maps than unguided AlphaFold3. In some cases, the guided structure outperforms PDB-deposited structure's faithfulness to the density (Table A3). Additionally, guided structures capture structural heterogeneity better than AlphaFold3 (Figures 2, 3). In some cases, guided structures capture the structural heterogeneity that PDB-deposited structures fail to model (Figure A1). Lastly, we are able to leverage the strong prior learned by AlphaFold3 to generate density-faithful ensembles in a fraction of the time required by conventional X-ray crystallography pipelines (Adams et al., 2010) (Table A5). In our opinion, this advancement not only improves the accuracy of computational structural modeling but also has the potential to automate workflows for crystallographers.

**Accelerated NMR ensemble structure determination.** Solution-state NMR enables the study of proteins in near-physiological aqueous environments, capturing conforma-

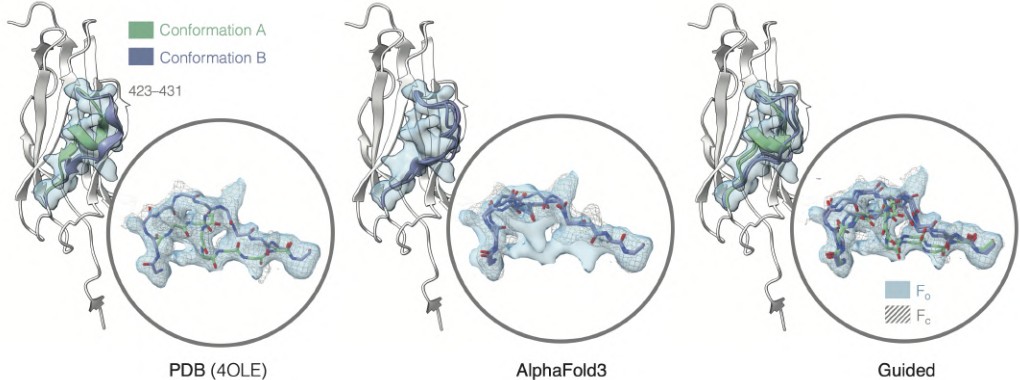

*Figure 3.* Crystallographic observation of the human NBR1 protein at $2.52\text{Å}$ resolution (PDB: `4OLE`) exhibits a multi-modal backbone distribution at `423-431` (conformation modes A and B color-coded in green and purple, respectively). AlphaFold3 predicts only conformation B while completely missing the helical conformation A. Electron density-guided AlphaFold3 predicts a bi-modally distributed ensemble better describing the observed electron density $F_o$. Light blue surfaces and gray meshes in the zoomed-in inserts depict the $0.3$ $[\text{e}^-/\text{Å}^3]$-isosurfaces of the observed and calculated electron densities, $F_o$ and $F_c$, respectively. Side chains in the inserts are omitted for clarity.

tional heterogeneity through nuclear interaction restraints such as nuclear Overhauser effects (NOEs) and scalar couplings (J-couplings). NMR-based structure determination typically employs restrained molecular dynamics (MD) simulations, requiring hundreds of independent trajectories to adequately sample conformational spaces consistent with experimental data—a computationally intensive process that struggles to balance accuracy, efficiency, and ensemble diversity (Lindorff-Larsen et al., 2005; Lange et al., 2008).

Here, we propose *NOE-guided AlphaFold3*, which refines AlphaFold-generated structural ensembles to satisfy NOE-derived distance restraints. The resulting ensembles adhere to experimental NOE data more faithfully than AlphaFold3 predictions and, *in some cases, even surpass the accuracy of existing PDB-deposited NMR ensembles* (see Table A4). In particular, we demonstrate that the ensembles produced by NOE-guided AlphaFold3 on ubiquitin, a benchmark system for NMR structure and dynamics, accurately capture experimentally observed conformational flexibility, as independently validated against experimentally-measured N-H $S^2$ order parameters (Figure 5; Lienin et al. (1998)). In contrast, standard AlphaFold3 predictions generate overly rigid ensembles inconsistent with ubiquitin's dynamic behavior. Finally, we note that our method dramatically *improves the NMR structure determination process from many hours to a few minutes, while retaining the accuracy obtained through MD*. We believe this will enable new experimental workflows for NMR structural biologists.

## 3. Protein structure inverse problems

**Notation.** We denote the amino-acid sequence of a protein as $\mathbf{a}$ and the corresponding 3D Cartesian coordinates of

all atoms as $\mathbf{X} = (\mathbf{x}_1, \ldots, \mathbf{x}_m)$, where $\mathbf{x}_i$ denotes the $i$-th atom in the structure. Note that $\mathbf{X}$ implicitly depends on $\mathbf{a}$ as the atom configuration is dependent on the amino acid identities.

**Problem statement.** Given a protein sequence $\mathbf{a}$ and an experimental observation $\mathbf{y}$, sample a *non-i.i.d.* ensemble of structures $\boldsymbol{\mathcal{X}} = \{\mathbf{X}^1, \ldots, \mathbf{X}^n\}$ from the posterior distribution $p(\boldsymbol{\mathcal{X}} \mid \mathbf{a}, \mathbf{y})$.

Using Bayes' rule, the posterior distribution can be factorized as, $p(\boldsymbol{\mathcal{X}} \mid \mathbf{a}, \mathbf{y}) \propto p(\mathbf{y} \mid \boldsymbol{\mathcal{X}}, \mathbf{a}) \cdot p(\boldsymbol{\mathcal{X}} \mid \mathbf{a})$, where $p(\mathbf{y} \mid \boldsymbol{\mathcal{X}}, \mathbf{a})$ is the *data term* representing the likelihood of the experimental observation given the structural ensemble and amino acid sequence. The knowledge of the instrument's forward model is embodied by the likelihood term. On the other hand, $p(\boldsymbol{\mathcal{X}} \mid \mathbf{a})$ is the *prior term* representing the probability of a structural ensemble given the amino acid sequence. A key distinction between these two terms is that the prior can be factorized into a product of independent priors for each sample in the ensemble $\boldsymbol{\mathcal{X}}$, whereas the likelihood is conditioned on the entire ensemble and hence is inseparable.

As the prior, we use AlphaFold3 (Abramson et al., 2024) to generate ensembles. To model the data term, we consider three distinct experimental modalities: crystallographic electron density maps (Section 3.1), nuclear Overhauser effect (NOE) restraints (Section 3.2), and sub-structure conditioning using known atom locations (Section 3.3).

### 3.1. Crystallographic electron densities

This section introduces the forward model of crystallographic electron density observables. Electron densities are volumetric images of the spatial charge distribution within

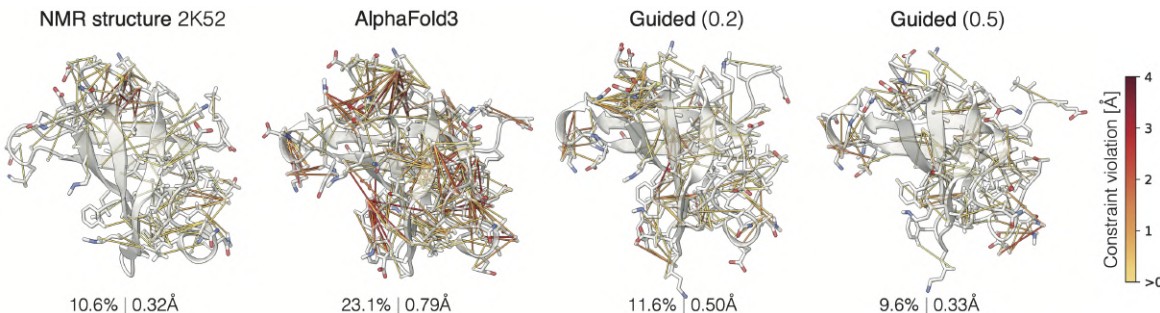

*Figure 4.* NOE constraint violations in the Methanocaldococcus jannaschii MJ1198 protein in the NMR structure ensemble (PDB: 2K52) and ensembles predicted by AlphaFold3 and using NOE-guidance with strength 0.2 and 0.5. Violated constraints are depicted as lines color-coded by the amount of violation. Percentage of violated constraints (out of total 1212) and their median violation are reported below each structure. A single best-fitting structure from each ensemble is shown for clarity.

the unit cell of a protein crystal lattice (Riley et al., 2021; van Zundert et al., 2018). The process of obtaining these maps begins with purifying and crystallizing the protein sample, followed by X-ray diffraction analysis (Smyth & Martin, 2000). When the crystal is exposed to an X-ray beam, it generates multiple diffraction patterns encoding the Fourier transform of the electron density function that is periodic on the crystal lattice (the Fourier transform is thus discrete on the reciprocal lattice). However, during this process phase information is lost, and a molecular replacement procedure is required to impute the missing phase angles, after which a 3D electron density distribution is reconstructed. The resulting electron density map reflects the average density of trillions of protein molecules in the crystal rather than that of an individual molecule.

We denote the observed electron density map as $F_o : \mathbb{R}^3 \to \mathbb{R}$ without explicitly distinguishing between the continuous map and its discretized version. Given $F_o$ and a protein structure ensemble $\mathcal{X}$, the log-likelihood $\log p(F_o \mid \mathcal{X}, \mathbf{a})$ quantifies the agreement between the experimental data and the electron density predicted by the ensemble. To compute this likelihood, we calculate the predicted electron density $F_c(\mathbf{X})$ for each structure $\mathbf{X}$ in the ensemble (see Appendix A.2, A.6 for details). The log-likelihood is given by

$$\log p(F_o \mid \mathcal{X}, \mathbf{a}) = -\left\| F_o - \frac{1}{n}\sum_{k=1}^{n} F_c(\mathbf{X}^k, \mathbf{a}) \right\|_1, \quad (1)$$

where we pragmatically choose the $L_1$ norm to quantify the discrepancy between the observed density and its calculated counterpart. Note that both $F_c$ and $F_c(\mathbf{X})$ are functions of the spatial coordinate $\boldsymbol{\xi}$ as explicated in the Appendix.

### 3.2. Nuclear Overhauser effect restraints

This section introduces the forward model for the nuclear Overhauser effect (NOE) restraints as measured using NMR spectroscopy. NOE restraints provide essential information about interatomic distances between biomolecules. The

NOE arises from through-space dipolar interactions between nuclear spins, usually involving hydrogen atoms. The interactions are also dependent on spatial proximity, with NOE effects observed typically when atoms are less than 6Å apart. NOE measurements represent an ensemble average over all conformations of the protein in solution, capturing its intrinsic structural heterogeneity.

Interatomic distances are usually measured in NOE spectroscopy (NOESY) experiments, which consist of a two-dimensional correlation spectrum. The proximity between two atoms is evidenced by a correlation peak at a position in the spectrum that correspond to the two resonance frequencies of the spins. The intensity of the NOE signals is proportional to the temporal average of the inverse sixth power of the interatomic distance and can be used to estimate spatial constraints within the protein. NMR signal intensities depend on several other factors, which obscures the dependency on the inverse sixth power of the distance to extent. While quantitative distances can be obtained (Vögeli, 2014), it is common to use the NOE signal intensity only semi-quantitatively. With some abuse of NMR physics, we henceforth assume that the distance average is observed directly, and define the NOE constraints as a set $D = \{(\underline{d}_{ij}, \overline{d}_{ij}) : (i,j) \in \mathcal{P}\}$ of pairs of lower and upper bounds on the ensemble average, $d_{ij}(\mathcal{X}) = \frac{1}{n}\sum_{k=1}^{n} d_{ij}(\mathbf{X}^k)$ of the distances $d_{ij}(\mathbf{X}^k) = \|\mathbf{x}_i^k - \mathbf{x}_j^k\|$ between pairs of atoms $i, j$ in individual structures $\mathbf{X}^k$. The log-likelihood is given by,

$$\log p(D \mid \mathcal{X}, \mathbf{a}) =$$
$$-\sum_{(i,j)\in\mathcal{P}} \left( \left[\underline{d}_{ij} - d_{ij}(\mathcal{X})\right]_+^2 + \left[d_{ij}(\mathcal{X}) - \overline{d}_{ij}\right]_+^2 \right), \quad (2)$$

where $[x]_+ = \max(x, 0)$. Additional computational details, noise model, and limitations are provided in App. A.2, A.12.

## 3.3. Substructure conditioning

In many cases, especially when refining crystallographic structures, it is useful to determine the protein structure only for a subset of amino acids while keeping the rest frozen. This can be achieved by a specialized likelihood term, that parallels the `SubstructureConditioner` in Chroma (Ingraham et al., 2023), which incorporates a reference structure to constrain and guide the optimization process during inference.

As the input, we will assume to be given a collection of reference atom locations $Y = \{\mathbf{y}_i : i \in A\}$ for some subset of atom indices $A$. Using a quadratic penalty on the deviation, the log-likelihood assumes the form

$$\log p(Y \mid \boldsymbol{\mathcal{X}}, \mathbf{a}) = -\frac{1}{n} \sum_{k=1}^{n} \sum_{i \in A} \|\mathbf{x}_i^k - \mathbf{y}_i\|^2 \quad (3)$$

Note that unlike the previously discussed forward models, this term is separable with respect to the individual ensemble constituents. See A.2 for the noise model.

# 4. Experiment-grounded AlphaFold3

## 4.1. Guiding AlphaFold3

One of the key features distinguishing AlphaFold3 from its predecessors (Jumper et al., 2021), is the introduction of a diffusion-based (Ho et al., 2020) generative model for protein structure prediction. This model acts as a prior over the all-atom distribution of protein structures and enforces structural coherence. AlphaFold3's forward diffusion process is modeled as a variance-preserving stochastic differential equation (SDE) (Song & Ermon, 2019; Weiss et al., 2023), whose backward SDE is a simplified variant of the formulation in (Karras et al., 2022),

$$d\mathbf{X} = -\left(\frac{1}{2}\mathbf{X} + \nabla_{\mathbf{X}} \log p_t(\mathbf{X} \mid \mathbf{a})\right)\beta_t dt + \sqrt{\beta_t}\mathbf{N}. \quad (4)$$

Here $\mathbf{X}$ containing the atomic coordinates acts as a diffusion variable, $\beta_t$ defines the noise schedule parameters, and $\mathbf{N} \sim \mathcal{N}(\mathbf{0}, \mathbf{I})$ is sampled from an isotropic normal distribution. The score function $\nabla_{\mathbf{X}} \log p_t(\mathbf{X} \mid \mathbf{a})$ is modeled using a atom-transformer based denoising network (Jumper et al., 2021; Abramson et al., 2024; Vaswani, 2017). AlphaFold3 samples random latent vectors from the base distribution $\mathbf{X}_T \sim \mathcal{N}(\mathbf{0}, \beta_0 \mathbf{I})$ and numerically integrate equation (4) from $t = T$ down to $t = 0$.

Here, rather than predicting a single structure, we propose to sample an ensemble $\boldsymbol{\mathcal{X}} = (\mathbf{X}^1, \ldots, \mathbf{X}^n)$ of $n$ structures. The SDE for single-structure generation in equation (4) can be straightforwardly generalized for ensemble sampling. In order to sample a *non-i.i.d* sample from the posterior distri-

bution, we further plug in the guidance score, obtaining,

$$d\begin{bmatrix} \mathbf{X}^1 \\ \vdots \\ \mathbf{X}^n \end{bmatrix} = -\left(\frac{1}{2}\begin{bmatrix} \mathbf{X}^1 \\ \vdots \\ \mathbf{X}^n \end{bmatrix} + \begin{bmatrix} \nabla_{\mathbf{X}^1} \log p_t(\mathbf{X}^1 \mid \mathbf{a}) \\ \vdots \\ \nabla_{\mathbf{X}^n} \log p_t(\mathbf{X}^n \mid \mathbf{a}) \end{bmatrix}\right)\beta dt$$
$$- \boldsymbol{\eta}\nabla_{\boldsymbol{\mathcal{X}}} \log p\left(\mathbf{y} \mid \mathbf{X}^1, \ldots, \mathbf{X}^n, \mathbf{a}\right)\beta dt + \sqrt{\beta_t}\begin{bmatrix} \mathbf{N}^1 \\ \vdots \\ \mathbf{N}^n \end{bmatrix},$$
$$(5)$$

where $\mathbf{N}^k \sim \mathcal{N}(\mathbf{0}, \mathbf{I})$. In the above equation, unconditional score term $\nabla_{\mathbf{X}^k} \log p_t(\mathbf{X}^k \mid \mathbf{a})$ is separable in the ensemble members. However, the guidance score term is not separable due to the *non-i.i.d* nature of the likelihood function. Additionally, the hyperparameter $\boldsymbol{\eta}$ can be used to scale the guidance score and direct the diffusion model to generate samples with high posterior likelihood. The pseudocode for guided AlphaFold3 and other implementation details are presented in Appendix A.1.

## 4.2. Force-field relaxation

After performing non-i.i.d. guided diffusion, we noticed that when sampling a large ensemble, the conformations tend to overfit to the noise in the experimental observations. This can significantly degrade the ensemble quality, often leading to artifacts such as broken bands and atomic clashes (Shapovalov & Dunbrack, 2011). To eliminate these artifacts, we remove from the ensembles structures with broken bonds (distance between any pair of bonded atoms exceeding the threshold $\tau_{\text{bond}}$ Å) or steric clashes (distance between any two atoms is less than $\tau_{\text{clash}}$ Å). Specific threshold values and implementation details are provided in Appendix A.7.

Post-filtering, subtle bond length violations and geometric inconsistencies may persist. To accurately model molecular interactions and eliminate geometric violations, we relax the structures with an off-the-shelf harmonic force-field, such as AMBER (Hornak et al., 2006). This ensures that all remaining structures are physically plausible with no structural artifacts while still maximizing the log-likelihood of the experimental observations.

## 4.3. Ensemble filtering using matching pursuit

Post relaxation, we employ a matching pursuit-based approach (Mallat & Zhang, 1993) to greedily select a subset of the relaxed ensemble, $\boldsymbol{\mathcal{X}}_{\mathcal{I}} = \{\mathbf{X}^k : k \in \mathcal{I}\}$, that best fits the observation $\mathbf{y}$. Starting with $\mathcal{I} = \emptyset$, during every iteration of the matching pursuit algorithm, we seek to maximize $\log p(\mathbf{y} \mid \boldsymbol{\mathcal{X}}_{\mathcal{I} \cup \{k\}}, \mathbf{a})$ over all $k \notin \mathcal{I}$. The optimal element $k$ is then added to the support set $\mathcal{I}$ and the process continues until the likelihood no longer increases or the maximum allowed ensemble size $n_{\text{max}}$ is reached. A detailed explanation of the matching pursuit-based ensemble

filtering procedure, along with its pseudocode, is provided in Appendix A.8.

## 5. Modeling crystallographic ensembles

In what follows, we investigate three distinct cases of conformational heterogeneity evident in X-ray crystal structures that are consistently mispredicted by the unguided AlphaFold3 and demonstrate that AlphaFold3 guidance with electron densities significantly improves these predictions.

**Structurally heterogeneous homologous proteins.** The first is the most obvious case where a specific protein is captured in different conformations over multiple experiments. Apart from different interactions with molecular partners, these altered conformations can result from differences in the expression or purification processes or different solvent conditions. As an illustration, we use the SARS-CoV-2 accessory protein that is encoded by the open reading frame ORF8 and facilitates immune evasion in infected host cells. This small (104 amino acids) protein has been structurally resolved in several independent works that demonstrate significant structural variability in the loop regions. PDB structures 7F5F and 7JX6 differing only by a single point mutation, crystallized individually without the presence of molecular partners and diffracting to the same resolution (1.6Å) exhibit major conformational variations in three loops as depicted in Figure 2. Due to essentially identical sequence conditioning, AlphaFold3 fails to capture both global conformations well, predicting a tight ensemble that is similar to one of the variants at one site and to the other at the other. Guiding the ensemble generation with the observed electron density allows to better capture the physical reality of these different structures. A quantitative evaluation of this example is presented in Table A1.

**Heterogeneity in equivalent environments.** The second case we explore is a more local and subtle version of the former, highlighting that the same amino acid segment in distinct but homologous proteins can adopt different conformations, even when the immediately surrounding protein environment is equivalent (and, thus, the local segments have identical contact networks). A key component of AlphaFold3 prediction is based on coevolutionary signals from multiple sequence alignment (MSA) of homologous proteins in order to discern contact maps (which, in the form of MSA embedding, condition the diffusion model as depicted in Figure 1). It is, therefore, unsurprising then that homologous proteins harboring regions of identical amino acid sequence embedded into an equivalent spatially adjacent amino acid environment are predicted to have identical local conformation even from distinct protein sequences. As an illustration, we show in Figure A1 a pair of such homologous proteins (PDB 4NE4 and 5TEU) featuring a sequence of 4 amino acids in two distinct turn conformations. While the

5TEU conformation variant is mispredicted by AlphaFold3, adding electron density guidance reproduces both variants faithfully. We also observe a more heterogeneous ensemble generated in the latter case, including flipped carbonyl oxygen and sidechain flexibility that better explain the density. A quantitative evaluation of this and additional homologous pairs is presented in Table A2.

**Heterogeneity within the same crystal.** In the last case we consider, the heterogeneous conformations are intrinsic to the protein itself and are observed in a single crystal measurement, with the electron density appearing markedly bi- or multi-modal. While the phenomenon is very common in flexible side chains, the heterogeneity in the backbone conformations has remained underappreciated. In fact, common visualization platforms (e.g., PyMOL (DeLano et al., 2002) and ChimeraX (Pettersen et al., 2021)) and structural modeling tools (e.g., GROMACS (Van Der Spoel et al., 2005)) frequently reading the first listed conformation as default and disregarding alternate conformations (Gutermuth et al., 2023). Nevertheless, the Protein Data Bank is actually riddled with such altlocs. A recent study by Rosenberg et al. (2024a) compiled a comprehensive catalog of alternate conformations from PDB structures and the same group showed that even for regions with well-separated and stable alternate conformations, structural ensemble predictors such as AlphaFold3 fail to reproduce the experimentally determined distributions or capture the bimodal nature of backbone conformations (Rosenberg et al., 2024b). Here we demonstrate that this same set of separated and stable conformations is well-modeled using electron density guidance. As an illustration, we use a protein decoded from NBR1, the neighbor of the human BRCA1 gene 1 implicated in breast cancer. The PDB structure 4OLE resolved to 2.52Å contains a region of 9 amino acids that was modeled as a superposition of two alternate conformations (Figure 3). While AlphaFold3 accurately predicts only one of the conformations, electron density guidance generates an ensemble capturing the bi-modal nature of the backbone and better explaining the density. A quantitative evaluation of this an other 11 cases is presented in Table A3. Figure A2 presents a quantitative evaluation of ensemble bi-modality (refer to Appendix A.10 for details) and demonstrates that density-guided AlphaFold3 consistently produces bi-modally distributed ensembles, while its unguided counterpart typically produces a single mode.

An extended analysis is presented in Table A9, with additional 15 proteins where regions up to 22 residues in some cases (5V2M & 6E2S) are optimized. In all cases, cosine similarity between observed and calculated electron density maps matched or exceeded that of the PDB entries. Similar results were observed for polypeptide chains.

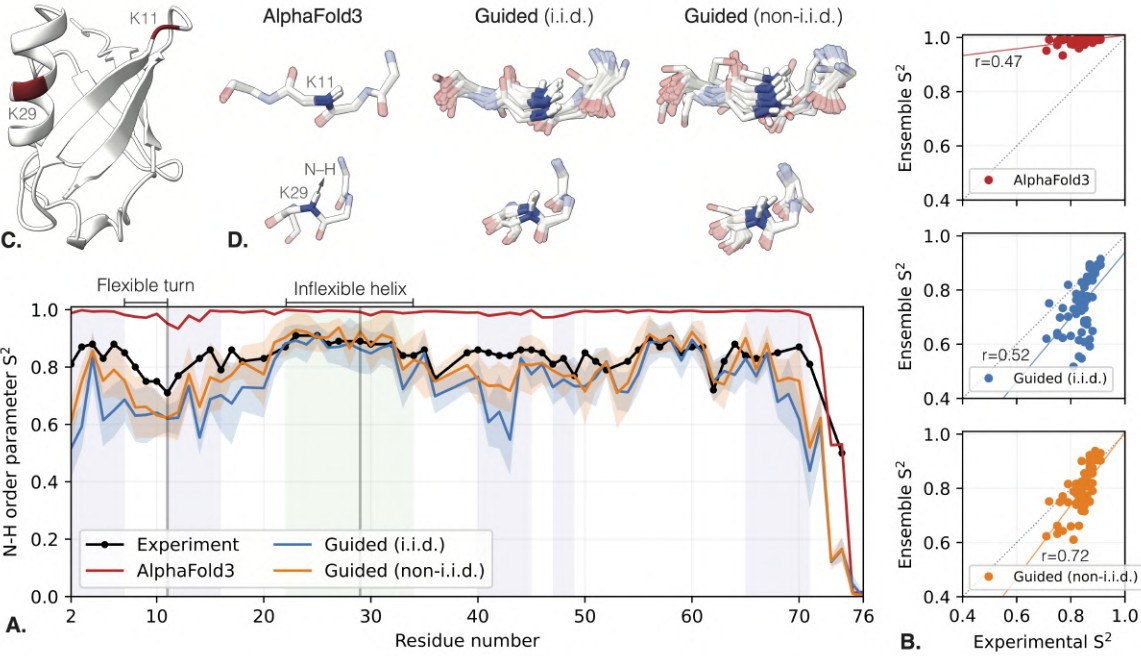

*Figure 5.* Comparison of conformational flexibility in ubiquitin ensembles predicted by AlphaFold3 and our i.i.d and non-i.i.d NOE-guided method. The N-H order parameter $S^2$ is calculated on the ensembles and compared to the experimental NMR observables. For guided predictions, plotted are the mean and standard deviation on 5 independent runs. Green and purple shadings indicate $\alpha$-helices and $\beta$-strands as predicted by DSSP (Frishman & Argos, 1995) (A). AlphaFold3 exhibits the lowest correlation to the experimental measurement ($r = 0.47$), while guiding AlphaFold3 with NOE measurements improves it to $r = 0.52$ for i.i.d guidance and $r = 0.72$ with non-i.i.d. ensembles (B). For two lysines at site 11 (entrance into the second $\beta$-strand from a flexible loop) and 29 (middle of $\alpha$-helix) highlighted in (C), guided ensembles exhibit more variability of the amide N-H bond direction in the flexible turn, while showing less variability in the inflexible helix, in agreement with the experimental observation (D).

# 6. Modeling NMR ensembles

NMR exploits the magnetic resonance of atomic nuclei to probe protein structure and dynamics. NOEs acquired from solution-state NMR, in particular, provide distances between atoms averaged over the ensemble of molecules in the sample and over time scales up to milliseconds. NOE-derived distances, thus, comprise the conformational heterogeneity.

**Status quo.** NMR structures are determined by integrating molecular dynamics (MD) simulations with NMR-derived restraints, using biomolecular force fields (Wang et al., 2004; Schwieters et al., 2006; Güntert, 2004). However, since NMR observables inherently reflect ensemble-averaged measurements, simulating single conformers often leads to mode collapse, producing rigid ensembles that poorly capture true conformational dynamics (Figure 4 in Lindorff-Larsen et al. (2005)). To address this, ensemble-based MD approaches—pioneered by (Lindorff-Larsen et al., 2005) and (Lange et al., 2008)—simulate multiple conformers simultaneously to satisfy experimental restraints. While effective, these methods remain computationally prohibitive, requiring days even for small systems like the 76-residue ubiquitin. While AlphaFold has revolutionized structure

prediction, its training on static X-ray crystallography data biases its ensembles toward rigid conformations, failing to capture conformational heterogeneity or satisfy NMR-derived restraints (Figure A7 and Table A4). This limits its utility for modeling protein dynamics. In what follows, we demonstrate that NOE-guided AlphaFold3 recovers ensembles that (i) rigorously obey NOE distance restraints; (ii) reproduce experimentally observed flexibility; and (iii) achieve this in minutes, overcoming the computational bottleneck of traditional ensemble methods.

**Ubiquitin**, widely regarded as the benchmark system for NMR-based protein dynamics studies, served as a critical test case for our method. To incorporate experimental constraints, we applied NOE-based guidance by integrating the likelihood term derived in Section 3.2 into the ensemble refinement framework defined by equation (5). This framework was explicitly parameterized with NOE-derived distance restraints obtained from the NMR structure PDB 1D3Z. NMR experiments based on $^{15}$N spin relaxation are a well-established way to assess protein dynamics (Palmer, 2004). Particularly, these experiments, which are entirely independent from NOE measurements, provide the amplitude

of motion of the amide bond vector on time scales shorter than a few nanoseconds. This amplitude is expressed as $(1-S^2)$, where $S^2$ is the so-called squared order parameter. This parameter is, thus, a convenient independent to validate the heterogeneity found in the ensembles determined by our NOE-guided AlphaFold approach. We computed the N-H $S^2$ bond order parameters and compared them against experimental measurements reported in (Lienin et al., 1998) (see Appendix A.14 for details). As illustrated in Figure 5, AlphaFold3 predictions yield ensembles dominated by rigid conformations, exhibiting only moderate correlation with the experimental $S^2$ ($r = 0.47$). In contrast, NOE-guided AlphaFold significantly improves agreement: i.i.d. guidance achieves $r = 0.52$ for i.i.d guidance, whereas *non-i.i.d.* ensemble sampling elevates it to $r = 0.72$. Notably, the refined ensembles better replicate the dynamic behavior observed in both flexible and structured regions, as evidenced by the distribution of N-H bond orientations across conformers.

**Peptides.** We used the benchmark from McDonald et al. (2023) from which we selected 20 peptides worst predicted by AlphaFold3, for which the NMR structures result violate less than $10\%$ distance restraints. This resulted in three structures, namely, 1DEC, 2LI3, 3BBG (Figure A7), for which AlphaFold3 produces a partially wrong fold than what is suggested by NMR measurements. This is visually depicted in Figure A7, and is evidenced by the violation of restraints of AlphaFold3 baseline presented in Table A4. We observed that NOE-guided AlphaFold3, both produces ensembles that obey $> 15\%$ more restraints and an order of magnitude lower error (see Table A4) and fixes the misprediction made by unguided AlphaFold3 (Figure A7).

**100 NMR spectra database.** We evaluated the proposed method on 9 proteins ($55 - 102$ residues) from a recently compiled database of NMR structures (Klukowski et al., 2024). While AlphaFold3 largely retains the correct fold for these systems, its baseline predictions exhibit, on average, $8\%$ more NOE restraint violations than the deposited NMR structures (Table A4). In all cases, NOE-guided AlphaFold reduced violations, and in half of the cases, the guided ensembles even outperformed PDB-deposited NMR structures in restraint compliance. Visual alignment with NMR ensembles (Figure A7) and quantitative agreement in $\rho$-RMSF (Table A4) confirm that guidance recovers conformational heterogeneity consistent with NMR structures. Increasing the guidance scale progressively enforces restraint satisfaction, as shown in Figure 4. Critically, these improvements are achieved within minutes (Table A6) – reaching the accuracy of MD-derived ensembles at a fraction of the computational cost.

A supplementary analysis is presented in Table A10, featuring 27 additional NMR proteins from NMRDb (Banfi & Patiny, 2008). For each protein, the guided ensemble better satisfies NOE restraints than unguided AlphaFold3 structures, and sometimes, even outperforms the corresponding NMR PDB structure.

## 7. Related Work

This work presents the first application of guided sampling with AlphaFold3 to resolve protein structural ensembles using crystallographic and NMR experimental data. Prior approaches have attempted similar goals but exhibit key limitations. Toth-Petroczy et al. (2016) uses a Potts model (Wu, 1982) to predict structured regions in intrinsically disordered proteins from evolutionary data. However, they do not leverage experimental measurements. Fadini et al. (2025) fit a single conformer to density maps by optimizing MSA contact maps but cannot model conformational heterogeneity. Liu et al. (2024) use str2str protein diffusion model (Lu et al., 2014) to generate cryo-EM–guided structural ensembles. While related, their focus is on a different modality. Maddipatla et al. (2024) fits ensembles to crystallographic density maps using Chroma (Ingraham et al., 2023) and captures multiple conformers to some extent. However, it fails to capture conformational heterogeneity over a long residue range due to Chroma's hierarchical formulation. Also, Chroma's sequence conditioning is only promoted, limiting enforcement of NMR distances. Finally, Levy et al. (2024) uses Chroma as their diffusion model and inherit similar limitations. Additionally, they use synthetic data instead of raw experimental observations.

## 8. Conclusion

In this paper, we present a general methodology for guiding AlphaFold3 using experimental observables and systematically evaluated its utility in fitting structural ensembles to crystallographic and NMR data. In crystallography, the robustness of our method demonstrated by the wide range of conditions of the evaluated targets: electron density maps covering a wide range of resolutions (from sub-Å to medium-low resolution) and qualities (covering high and low B-factors), protein segments with different lengths, variable amino acid identities, different secondary structure contexts, and AlphaFold3 mispredictions of severity ranging from the subtle conformation of the carbonyl oxygen to loop regions predicted tens of Ångstroms away from the experimentally observed location. Likewise, in NMR we show a great variability of the number and quality of NOE constraints in the evaluated structures. In the future, we intend to generalize the method to jointly sampling ensembles representing multiple molecules to model protein complexes and protein-ligand interactions. We also plan to extend the model to single- and multi-particle cryoEM where handling conformational heterogeneity constitutes a major challenge.

## Acknowledgments

This work was supported by the Israeli Science Foundation (ISF) grant number $1834/24$. We acknowledge support from the Austrian Science Fund (FWF, grant numbers I5812-B and I6223) and the financial support of the Helmsley Fellowships Program for Sustainability and Health. This research uses resources of the Institute of Science and Technology Austria's scientific computing cluster.

## Impact statement

The work presented here attempts to advance the modeling of protein structure and dynamics using experiment-guided AlphaFold3. The developed methods lead to solving existing tasks in structural biology significantly faster and more accurately. They may further permit the utilization of the wealth of experimental structural and dynamic measurements currently unused for the generative modeling of protein ensembles. Since proteins are fundamental ingredients of life and are implicated in health and disease, the potential impact on basic and applied research may be profound. We do not see any special ethical concerns worth highlighting.

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

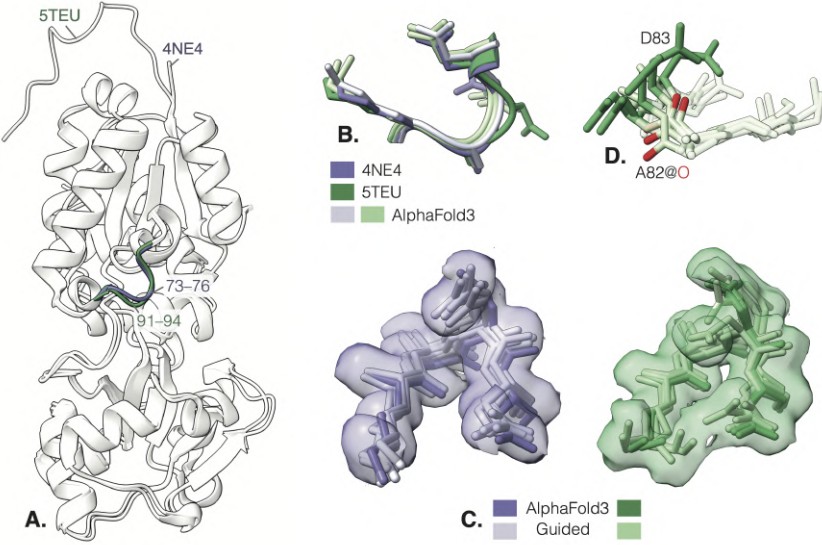

*Figure A1.* Crystallographic observations of a pair of homologous proteins (PDB: 4NE4 at 1.73Å resolution and 5TEU at 1.62Å). The pair features a distinct turn conformation at a corresponding site in residues 73-76 in 4NE4 and 91-94 in 5TEU despite identical local amino acid sequences and contacts with the environment (A). AlphaFold3 predicts the conformation of 4NE4 in both sequences mispredicting the conformation of 5TEU (B). Electron density-guided AlphaFold3 corrects these predictions by producing ensembles fitting well into the observed electron densities $F_o$ of both structures depicted as the $0.3 \, [e^-/Å^3]$-isosurfaces (C). The ensemble predicted for 5TEU exhibits conformation heterogeneity with two flipped states of the carbonyl oxygen in A82 (highlighted in red) and a flexible side chain in A82 (dark green) better explaining the $F_o$ (D).

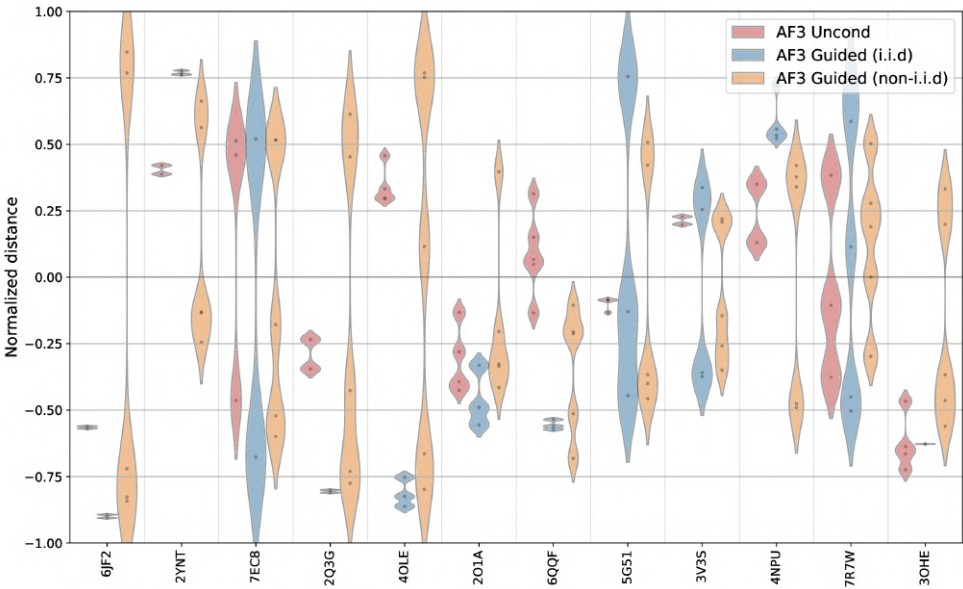

*Figure A2.* Distribution of normalized distances to conformations A ($-1$) and B ($+1$) in generated ensembles for 12 protein structures. Compared are ensembles generated by AlphaFold3 and our electron density i.i.d. and non-i.i.d. guidance. While unguided AlphaFold3 typically fails to capture the multi-modal nature of these structures, non-i.i.d. consistently produces multi-modal ensembles.

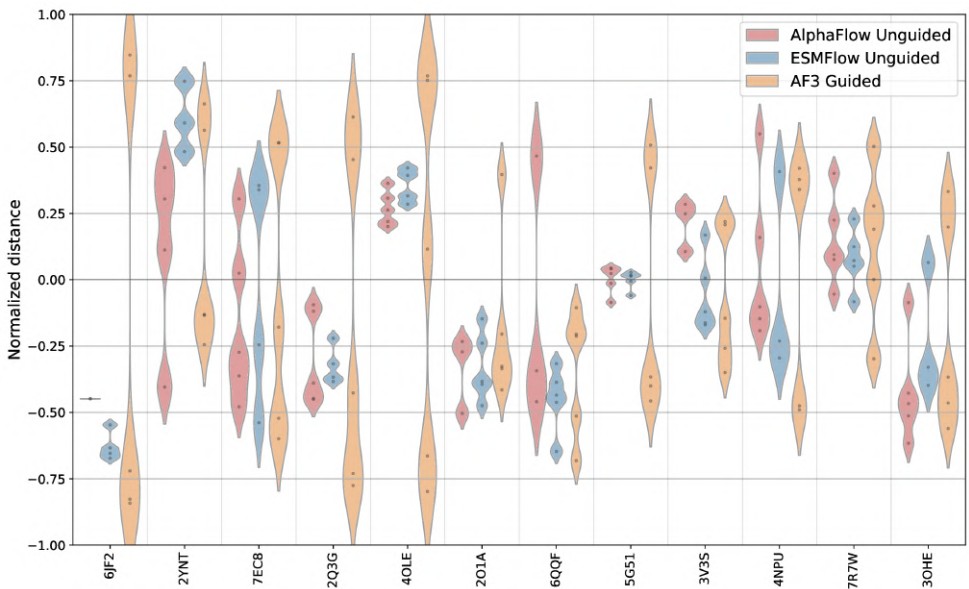

*Figure A3.* Distribution of normalized distances to conformations A (−1) and B (+1) in generated ensembles for 12 protein structures. Compared are ensembles generated by Alphaflow, ESMFlow (Jing et al., 2024) and our electron density non-i.i.d. guidance. While Alphaflow and ESMFlow typically fail to capture the multi-modal nature of these structures, non-i.i.d. consistently produces multi-modal ensembles.

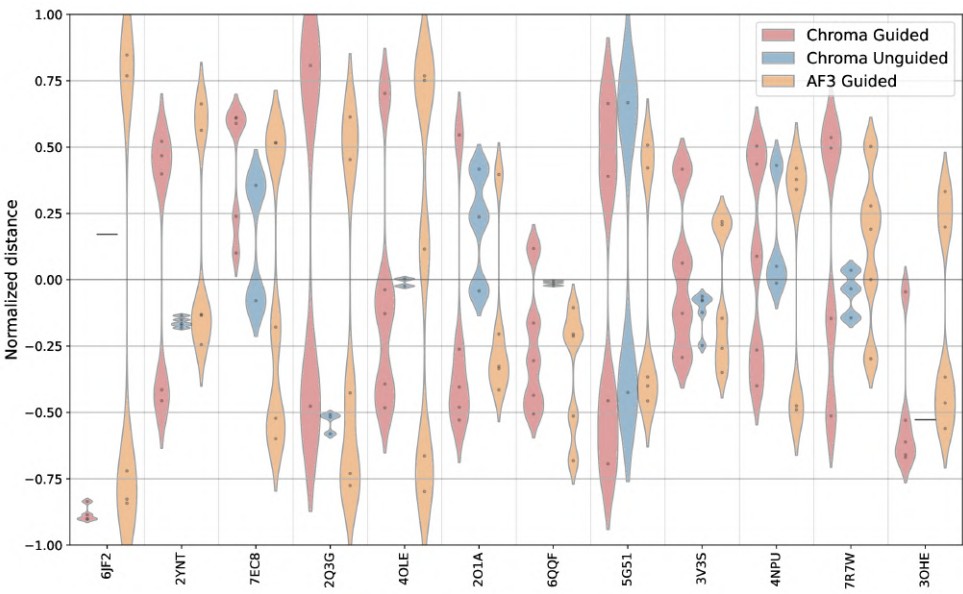

*Figure A4.* Distribution of normalized distances to conformations A (−1) and B (+1) in generated ensembles for 12 protein structures. Compared are ensembles generated by Chroma (Ingraham et al., 2023), non-i.i.d. guided Chroma (Maddipatla et al., 2024) and our electron density non-i.i.d. guidance. While non-i.i.d. guided Chroma captures multimodal distributions to some extent, our approach achieves more consistent bimodal across different proteins.

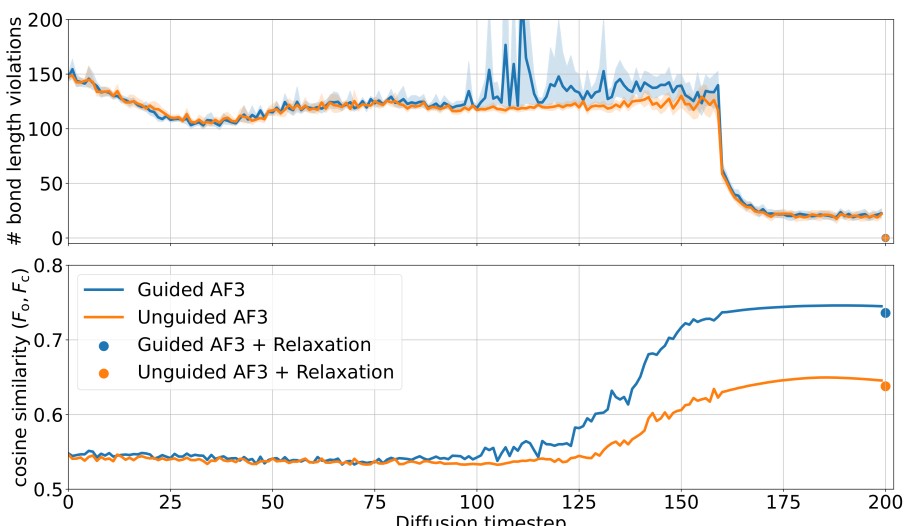

*Figure A5.* Quantitative assessment of generated structure validity (top plot) and agreement with the experimental observation (bottom plot) during the progress of diffusion iterations using an electron density likelihood for guidance. Structural validity is quantified as the number of violated bond lengths, while the agreement to experiment is measured using the cosine similarity between $F_{\mathrm{o}}$ and $F_{\mathrm{c}}$.

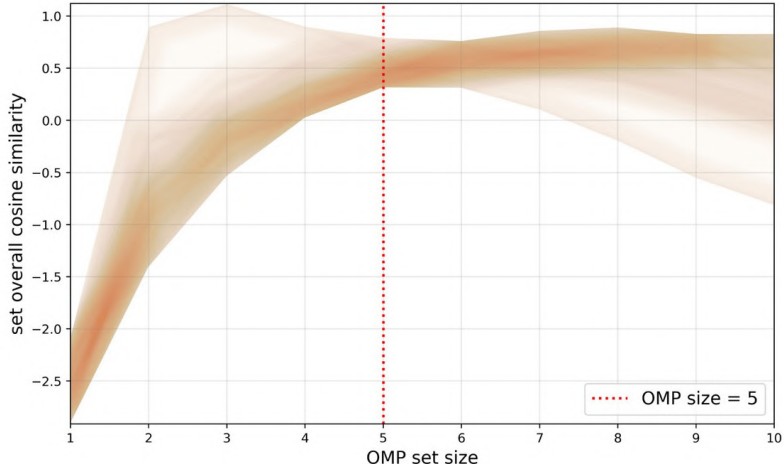

*Figure A6.* Ablation study depicting faithfulness to $F_{\mathrm{o}}$ as we add more samples to the chosen ensemble $\mathcal{X}$. Faithfulness to the experimental observation is measured using the normalized cosine similarity between $F_{\mathrm{o}}$ and $F_{\mathrm{c}}$.

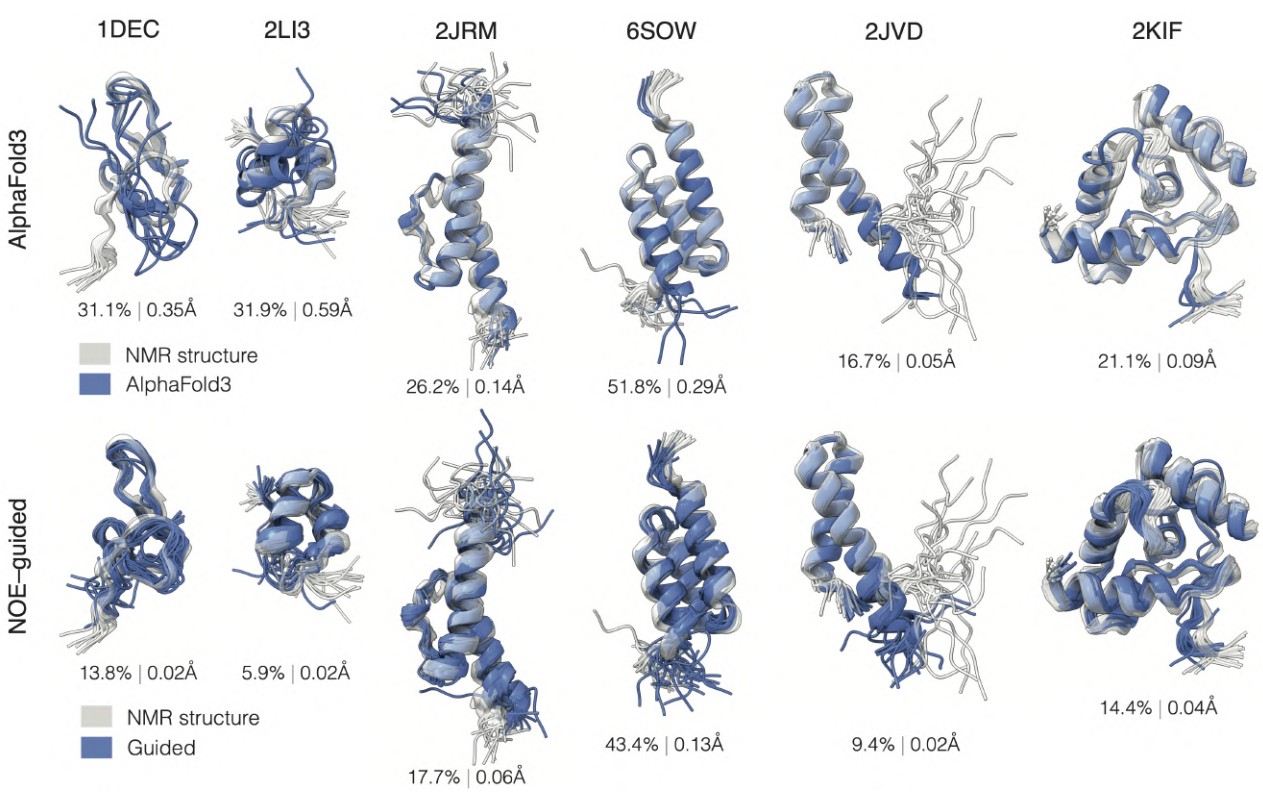

*Figure A7.* Conformation ensembles generated for six proteins using AlphaFold3 (first row) and the proposed NOE guidance (second row). Ensembles are visualized in blue overlaid on corresponding NMR structures solved from the same NOESY data. PDB identifiers are indicated above each structure. The numbers below report the percentage of violated constraints and the median violation.

| PDB ID | Residue region | Region sequence | Resolution (Å) | PDB | AlphaFold3 | Guided (non-i.i.d.) |
|--------|---------------|-----------------|----------------|-----|-----------|---------------------|
| 7F5F:A | 102 − 112 | CSFYEDFLEYH | 1.62 | 0.717 | 0.562 | **0.699** |
| 7JX6:A | 102 − 112 | | 1.61 | 0.698 | 0.561 | **0.696** |
| 7F5F:A | 38 − 41 | PIHF | 1.62 | 0.788 | 0.645 | **0.785** |
| 7JX6:A | 38 − 41 | | 1.61 | 0.739 | 0.579 | **0.736** |
| 7F5F:A | 72 − 78 | QYIDIGN | 1.62 | 0.668 | 0.660 | **0.736** |
| 7JX6:A | 72 − 78 | | 1.61 | 0.652 | 0.552 | **0.657** |

*Table A1.* Quantitative evaluation of cosine similarity between the observed and calculated electron density maps (higher is better) on three structurally dissimilar crystallographic structures of the SARS-CoV-2 ORF8 protein. Colored in **green** are the cases in which the ensemble produced by the density-guided AlphaFold3 better fits the observed electron density better than the unguided counterpart, and in **blue** the cases in which the generated ensemble also outperforms the structure deposited in the PDB.

| PDB ID | Residue region | Region sequence | Resolution [Å] | PDB | AlphaFold3 | Guided (non-i.i.d.) |
|--------|---------------|-----------------|----------------|-----|-----------|---------------------|
| 4QTD:A | 225 − 228 | FPGR | 1.5 | 0.825 | 0.666 | **0.810** |
| 7KSI:A | 263 − 266 | | 1.75 | 0.831 | 0.627 | **0.819** |
| 4QTD:A | 280 − 291 | FPADSEHNKLKA | 1.5 | 0.792 | 0.626 | **0.771** |
| 7KSI:A | 318 − 329 | | 1.75 | 0.783 | 0.575 | **0.780** |
| 5LL7:A | 238 − 241 | CGVY | 1.40 | 0.855 | 0.754 | **0.830** |
| 3DW0:A | 283 − 286 | | 1.60 | 0.767 | 0.625 | **0.749** |
| 2D7C:A | 110 − 113 | RDHA | 1.75 | 0.782 | 0.685 | **0.785** |
| 2F9L:A | 110 − 113 | | 1.55 | 0.701 | 0.693 | **0.739** |
| 3F1L:A | 21 − 24 | SDGI | 0.95 | 0.886 | 0.744 | **0.859** |
| 3G1T:A | 21 − 24 | | 1.70 | 0.859 | 0.810 | **0.850** |
| 5XNE:A | 211 − 214 | EDCT | 1.50 | 0.820 | 0.775 | **0.821** |
| 6J3D:A | 211 − 214 | | 1.70 | 0.849 | 0.798 | **0.841** |
| 4NE4:A | 73 − 76 | ADDP | 1.73 | 0.734 | 0.628 | **0.728** |
| 5TEU:A | 91 − 94 | | 1.62 | 0.754 | 0.666 | **0.763** |
| 2ESK:A | 17 − 20 | PPAQ | 1.36 | 0.699 | 0.626 | **0.709** |
| 1Z2U:A | 17 − 20 | | 1.10 | 0.751 | 0.561 | **0.728** |
| 2ESK:A | 26 − 29 | VGDD | 1.36 | 0.710 | 0.648 | **0.696** |
| 1Z2U:A | 26 − 29 | | 1.10 | 0.761 | 0.611 | **0.745** |
| 2ESK:A | 113 − 119 | PNPDDPL | 1.36 | 0.774 | 0.712 | **0.777** |
| 1Z2U:A | 113 − 119 | | 1.10 | 0.799 | 0.635 | **0.778** |
| 2IE8:A | 321 − 324 | VPPF | 1.80 | 0.654 | 0.404 | **0.560** |
| 1V6S:A | 321 − 324 | | 1.50 | 0.864 | 0.518 | **0.851** |
| 2IE8:A | 284 − 287 | PVPY | 1.80 | 0.697 | 0.483 | **0.671** |
| 1V6S:A | 284 − 287 | | 1.50 | 0.812 | 0.540 | **0.817** |
| 2IE8:A | 354 − 363 | VNRLGLKERF | 1.80 | 0.639 | 0.454 | **0.640** |
| 1V6S:A | 354 − 363 | | 1.50 | 0.725 | 0.487 | **0.731** |

*Table A2.* Quantitative evaluation of cosine similarity between the observed and calculated electron density maps (higher is better) on homologous protein pairs harboring locally identical amino acid sequences in the identical environment (same contacts). Colored in **green** are the cases in which the ensemble produced by the density-guided AlphaFold3 better fits the observed electron density better than the unguided counterpart, and in **blue** the cases in which the generated ensemble also outperforms the structure deposited in the PDB.

| PDB ID | Residue region | Region sequence | Resolution [Å] | PDB | AlphaFold3 | Guided AlphaFold3 Non-i.i.d. | i.i.d. |
|---|---|---|---|---|---|---|---|
| 3OHE:A | $98 - 103$ | YQGDPAW | 1.20 | 0.773 | 0.706 | **0.761** | 0.755 |
| 2YNT:A | $183 - 185$ | GNG | 1.60 | 0.750 | 0.702 | **0.745** | 0.733 |
| 2Q3G:A | $24 - 27$ | FNVP | 1.11 | 0.755 | 0.687 | **0.737** | 0.716 |
| 3V3S:B | $245 - 250$ | KAQERD | 1.90 | 0.798 | 0.736 | 0.800 | **0.804** |
| 7EC8:A | $187 - 190$ | DGGI | 1.35 | 0.878 | 0.839 | **0.860** | 0.855 |
| 2O1A:A | $50 - 53$ | KQNN | 1.60 | 0.759 | 0.739 | **0.757** | 0.746 |
| 5G51:A | $290 - 295$ | GSASDQ | 1.45 | 0.748 | 0.447 | **0.749** | 0.742 |
| 4OLE:B | $423 - 431$ | STEKKDVLV | 2.52 | 0.880 | 0.809 | **0.882** | 0.854 |
| 6JF2:A | $129 - 133$ | VTAGG | 2.00 | 0.862 | 0.752 | **0.857** | 0.845 |
| 4NPU:B | $133 - 136$ | FEEI | 1.50 | 0.795 | 0.730 | **0.797** | 0.784 |
| 7R7W:B | $46 - 50$ | IEKVE | 1.17 | 0.760 | 0.718 | **0.761** | 0.748 |
| 6QQF:A | $68 - 75$ | RTPGSRNL | 1.95 | 0.833 | 0.814 | **0.832** | 0.825 |

*Table A3.* Quantitative evaluation of cosine similarity between the observed and calculated electron density maps (higher is better) on structures with separated multi-modal backbone conformations (altlocs) from (Rosenberg et al., 2024a). Colored in **green** are the cases in which the ensemble produced by the density-guided AlphaFold3 better fits the observed electron density better than the unguided counterpart, and in **blue** the cases in which the generated ensemble also outperforms the structure deposited in the PDB.

| PDB ID | Source | len | #NOEs | NMR (PDB) Viol.% | Viol.Å | AlphaFold3 Viol.% | Viol.Å | $\rho$-RMSF | Guided AlphaFold3 Viol.% | Viol.Å | $\rho$-RMSF |
|---|---|---|---|---|---|---|---|---|---|---|---|
| 1DEC | pept. | 39 | 602 | 11.4% | 0.019 | 31.2% | 0.673 | 0.80 | **15.0%** | 0.062 | **0.86** |
| 2LI3 | pept. | 30 | 354 | 1.0% | 0.003 | 29.3% | 1.104 | 0.57 | **5.3%** | 0.023 | **0.66** |
| 3BBG | pept. | 40 | 535 | 3.5% | 0.023 | 33.3% | 1.253 | 0.63 | 21.7% | 0.174 | 0.49 |
| 1YEZ | NMRDb | 68 | 1512 | 11.4% | 0.074 | 12.4% | 0.097 | 0.76 | **7.8%** | **0.046** | **0.79** |
| 2JRM | NMRDb | 60 | 2706 | 12.2% | 0.069 | 19.7% | 0.168 | 0.83 | 13.0% | 0.072 | **0.86** |
| 2JVD | NMRDb | 48 | 1324 | 6.5% | 0.024 | 13.1% | 0.079 | 0.60 | 7.9% | 0.032 | **0.70** |
| 2K52 | NMRDb | 74 | 1212 | 14.9% | 0.092 | 27.6% | 0.295 | 0.60 | **14.3%** | **0.060** | **0.80** |
| 2K57 | NMRDb | 55 | 1200 | 8.9% | 0.070 | 17.8% | 0.135 | 0.70 | 10.1% | **0.058** | **0.89** |
| 2KIF | NMRDb | 102 | 3124 | 11.8% | 0.077 | 19.3% | 0.168 | 0.57 | 13.3% | 0.081 | **0.91** |
| 2KRS | NMRDb | 74 | 1305 | 19.8% | 0.145 | 13.8% | 0.117 | 0.82 | **10.3%** | **0.057** | **0.83** |
| 2MA6 | NMRDb | 61 | 1077 | 10.4% | 0.070 | 9.7% | 0.076 | 0.93 | **8.4%** | **0.053** | **0.89** |
| 6SOW | NMRDb | 58 | 1589 | 31.4% | 0.201 | 42.6% | 0.422 | 0.30 | 36.5% | 0.197 | **0.54** |

*Table A4.* Quantitative evaluation of restraint violation and backbone flexibility for protein structures sourced from the 100 NMR spectra database (NMRDb; Klukowski et al. (2024)) and benchmark peptides (pept., McDonald et al. (2023)). Violation percentages (**Viol. %**) quantify the fraction of experimental NOE restraints that are not satisfied, while violation distances (**Viol. Å**) report by how far the modeled ensemble deviates from these restraints. Correlation with ground-truth RMSF ($\rho$-**RMSF**) measures the accuracy of the ensemble's backbone flexibility. Colored in **blue** are cases in which the ensemble produced by NOE-guided AlphaFold3 better satisfies the distance restraints than the corresponding NMR structure deposited in the PDB. Colored in **green** are the cases in which the NOE-guided AlphaFold3 achieves at least 15% fewer restraints than AlphaFold3. Colored in **black** are cases in which NOE-guided AlphaFold3better agrees with the NMR structure in terms of conformational flexibility compared to AlphaFold3.

| PDB ID | Residue range length | Seq. length | Runtime (seconds) | |
|---|---|---|---|---|
| | | | Guided AlphaFold3 | AlphaFold3 |
| 6JF2 | 3 | 145 | 99 | 31 |
| 2YNT | 3 | 221 | 183 | 43 |
| 7EC8 | 3 | 265 | 225 | 52 |
| 4OLE | 9 | 120 | 73 | 25 |
| 2B3P | 7 | 228 | 181 | 45 |
| 6E2S | 20 | 311 | 324 | 82 |
| 5SUJ | 4 | 380 | 427 | 96 |
| 5V2M | 20 | 311 | 311 | 60 |

*Table A5.* Runtime comparison (in seconds) for Non-I.I.D. crystallographic guidance with AlphaFold3 versus unguided AlphaFold3 across multiple PDB entries.

| Ensemble size | Runtime (seconds) |
|---|---|
| 16 | 503 |
| 32 | 847 |
| 64 | 1545 |
| 128 | 2978 |

*Table A6.* Runtime comparison (in seconds) of performing Non-I.I.D. NOE guidance with AlphaFold3 across different ensemble sizes.

| PDB ID | #NOEs | NMR (PDB) | | AlphaFlow | | ESMFlow | | Guided AlphaFold3 | |
|---|---|---|---|---|---|---|---|---|---|
| | | Viol.% | Viol.Å | Viol.% | Viol.Å | Viol.% | Viol.Å | Viol.% | Viol.Å |
| 1DEC | 602 | 11.4% | 0.019 | 48.7% | 1.687 | 40.2% | 0.863 | **15.0**% | 0.062 |
| 2LI3 | 354 | 1.0% | 0.003 | 28.8% | 1.150 | 29.2% | 1.274 | **5.3**% | 0.023 |
| 3BBG | 535 | 3.5% | 0.023 | 34.8% | 0.654 | 23.5% | 0.169 | 21.7% | 0.174 |
| 1YEZ | 1512 | 11.4% | 0.074 | 12.4% | 0.092 | 12.3% | 0.075 | **7.8**% | **0.046** |
| 2JRM | 2706 | 12.2% | 0.069 | 18.2% | 0.162 | 17.9% | 0.165 | 13.0% | 0.072 |
| 2JVD | 1324 | 6.5% | 0.024 | 17.8% | 0.117 | 18.4% | 0.129 | 7.9% | 0.032 |
| 2K52 | 1212 | 14.9% | 0.092 | 37.6% | 0.471 | 35.8% | 0.427 | **14.3**% | **0.060** |
| 2K57 | 1200 | 8.9% | 0.070 | 19.3% | 0.152 | 19.9% | 0.137 | 10.1% | **0.058** |
| 2K1F | 3124 | 11.8% | 0.077 | 21.9% | 0.147 | 21.3% | 0.143 | 13.3% | 0.081 |
| 2KRS | 1305 | 19.8% | 0.145 | 18.1% | 0.127 | 15.6% | 0.097 | **10.3**% | **0.057** |
| 2MA6 | 1077 | 10.4% | 0.070 | 18.0% | 0.156 | 18.9% | 0.173 | **8.4**% | **0.053** |
| 6S0W | 1589 | 31.4% | 0.201 | 47.5% | 0.460 | 44.1% | 0.401 | 36.5% | 0.197 |

*Table A7.* Table A4 extended with more baselines including AlphaFlow and ESMFlow. Colored in **blue** are cases in which the ensemble produced by NOE-guided AlphaFold better satisfies the distance restraints than the corresponding NMR structure deposited in the PDB. Colored in **green** are the cases in which the NOE-guided AlphaFold achieves at least 15% fewer restraints than AlphaFold.

| PDB ID | PDB | Guided Chroma | ESMFlow | AlphaFlow | Guided AlphaFold3 |
|---|---|---|---|---|---|
| 3OHE:A | 0.773 | 0.730 | 0.674 | 0.711 | **0.761** |
| 2YNT:A | 0.759 | 0.740 | 0.725 | 0.714 | **0.745** |
| 2Q3G:A | 0.755 | 0.743 | 0.646 | 0.642 | **0.737** |
| 3V3S:A | 0.798 | 0.774 | 0.749 | 0.698 | **0.800** |
| 7EC8:A | 0.878 | 0.869 | 0.841 | 0.829 | **0.860** |
| 2O1A:A | 0.759 | 0.754 | 0.736 | 0.737 | **0.757** |
| 5G51:A | 0.748 | 0.749 | 0.486 | 0.513 | **0.749** |
| 4OLE:B | 0.880 | 0.863 | 0.792 | 0.793 | **0.882** |
| 6JF2:A | 0.833 | 0.848 | 0.795 | 0.735 | **0.832** |
| 4NPU:B | 0.796 | 0.795 | 0.756 | 0.772 | **0.797** |
| 7R7W:B | 0.760 | 0.758 | 0.692 | 0.698 | **0.761** |
| 6QQF:A | 0.833 | 0.832 | 0.773 | 0.775 | **0.832** |

*Table A8.* Table A3 extended with baselines. Colored in **green** are the cases in which the ensemble produced by the density-guided AlphaFold3 better fits the observed electron density better than other baselines (barring the PDB structure), and in **blue** the cases in which the generated ensemble also outperforms the structure deposited in the PDB.

| PDB ID | Residue Range | PDB | AlphaFold3 | AlphaFlow | ESMFlow | Guided AlphaFold3 |
|---|---|---|---|---|---|---|
| 2B3P:A | 189 − 195 | 0.674 | 0.534 | 0.568 | 0.566 | **0.697** |
| 2IZR:A | 208 − 213 | 0.853 | 0.745 | 0.751 | 0.782 | **0.853** |
| 3UGC:A | 1050 − 1053 | 0.781 | 0.695 | 0.723 | 0.724 | **0.770** |
| 5SUJ:B | 198 − 201 | 0.753 | 0.551 | 0.509 | 0.539 | **0.720** |
| 8A4A:A | 41 − 49 | 0.623 | 0.575 | **0.643** | 0.585 | 0.592 |
| 5V2M:A | 293 − 312 | 0.672 | 0.643 | 0.654 | 0.573 | **0.690** |
| 7RYL:A | 225 − 232 | 0.689 | 0.648 | 0.655 | 0.650 | **0.665** |
| 6RHT:A | 200 − 218 | 0.785 | 0.713 | 0.716 | 0.709 | **0.760** |
| 7A7M:A | 189 − 195 | 0.806 | 0.761 | 0.777 | 0.783 | **0.791** |
| 5JUD:A | 251 − 263 | 0.808 | 0.762 | 0.789 | **0.802** | 0.797 |
| 6E2S:A | 292 − 313 | 0.709 | 0.683 | 0.633 | 0.629 | **0.742** |
| 1AWR:G | 1 − 6 | 0.442 | 0.360 | 0.344 | 0.310 | **0.436** |
| 2DF6:C | 1 − 18 | 0.515 | 0.364 | 0.404 | 0.320 | **0.481** |
| 6I42:B | 1 − 13 | 0.506 | 0.430 | 0.456 | 0.418 | **0.491** |
| 7ABT:B | 1 − 8 | 0.522 | 0.403 | 0.359 | 0.368 | **0.504** |

*Table A9.* Quantitative evaluation of cosine similarity between the observed and calculated electron density maps (higher is better) on an expanded set of proteins, including those with extended alternative conformations (altloc regions) and polypeptides (second half of table). Colored in **green** are the cases in which the ensemble produced by the density-guided AlphaFold3 better fits the observed electron density better than the unguided counterpart, and in **blue** the cases in which the generated ensemble also outperforms the structure deposited in the PDB.

| PDB ID | Seq. Len | #NOEs | NMR (PDB) | | AlphaFold3 | | Guided AlphaFold3 | |
|--------|----------|-------|-----------|--------|------------|--------|-------------------|--------|
| | | | Viol.% | Viol.Å | Viol.% | Viol.Å | Viol.% | Viol.Å |
| 1PQX | 91 | 1395 | 0.93% | 0.0003 | 5.30% | 0.093 | 4.44% | 0.045 |
| 2JRM | 60 | 993 | 10.27% | 0.052 | 19.44% | 0.184 | 16.62% | 0.133 |
| 2JT1 | 71 | 1202 | 14.23% | 0.057 | 19.97% | 0.132 | 15.64% | 0.072 |
| 2JVD | 48 | 1029 | 9.91% | 0.037 | 16.23% | 0.103 | 12.63% | 0.048 |
| 2K0M | 104 | 1834 | 12.76% | 0.059 | 19.79% | 0.177 | 15.87% | 0.112 |
| 2K3A | 155 | 1872 | 18.59% | 0.101 | 21.53% | 0.162 | 16.08% | 0.080 |
| 2K3D | 87 | 291 | 33.68% | 1.525 | 5.50% | 0.013 | 4.47% | 0.009 |
| 2K5D | 110 | 1615 | 14.92% | 0.067 | 20.43% | 0.208 | 16.41% | 0.121 |
| 2K75 | 106 | 1227 | 15.40% | 0.068 | 14.75% | 0.097 | 11.08% | 0.049 |
| 2KD1 | 118 | 2142 | 15.50% | 0.069 | 21.66% | 0.181 | 18.07% | 0.120 |
| 2KJR | 95 | 1281 | 21.70% | 0.101 | 26.54% | 0.229 | 20.61% | 0.114 |
| 2KKZ | 134 | 264 | 34.09% | 0.856 | 9.85% | 0.057 | 9.85% | 0.047 |
| 2KZV | 92 | 808 | 15.47% | 0.083 | 18.44% | 0.166 | 12.75% | 0.075 |
| 2L33 | 91 | 2310 | 15.19% | 0.066 | 23.46% | 0.200 | 20.30% | 0.129 |
| 2L82 | 162 | 4582 | 16.54% | 0.078 | 36.77% | 0.719 | 25.56% | 0.170 |
| 2L8V | 143 | 1167 | 19.71% | 0.090 | 23.74% | 0.188 | 18.85% | 0.092 |
| 2LF2 | 175 | 1825 | 18.25% | 0.091 | 19.62% | 0.159 | 15.89% | 0.081 |
| 2LGH | 144 | 1479 | 14.27% | 0.053 | 18.80% | 0.132 | 14.13% | 0.045 |
| 2LK2 | 89 | 845 | 13.37% | 0.063 | 14.56% | 0.107 | 11.72% | 0.044 |
| 2LML | 87 | 1038 | 13.58% | 0.061 | 14.16% | 0.088 | 11.18% | 0.038 |
| 2LTL | 119 | 2391 | 10.96% | 0.050 | 18.32% | 0.154 | 12.80% | 0.075 |
| 2LTM | 107 | 2598 | 13.32% | 0.059 | 21.29% | 0.166 | 15.94% | 0.088 |
| 2LX7 | 60 | 407 | 6.88% | 0.029 | 7.86% | 0.061 | 6.14% | 0.022 |
| 2MA6 | 61 | 716 | 10.75% | 0.057 | 15.64% | 0.151 | 11.45% | 0.057 |
| 2MK2 | 109 | 1268 | 10.02% | 0.043 | 14.43% | 0.132 | 9.07% | 0.050 |
| 2MQL | 105 | 48 | 22.92% | 0.818 | 25.00% | 0.788 | 22.92% | 0.609 |
| 6F3K | 353 | 109 | 43.12% | 0.700 | 38.53% | 0.624 | 13.76% | 0.046 |

*Table A10.* Quantitative evaluation of restraint violation and backbone flexibility on an expanded set of proteins from NMRDb. Violation percentages (**Viol. %**) quantify the fraction of experimental NOE restraints that are not satisfied, while violation distances (**Viol. Å**) report by how far the modeled ensemble deviates from these restraints. Colored in **blue** are cases in which the ensemble produced by NOE-guided AlphaFold3 better satisfies the distance restraints than the corresponding NMR structure deposited in the PDB. Colored in **green** are the cases where the ensemble abides by NOEs better than counterparts (excluding PDB).

---

**Algorithm 1** AlphaFold3 guidance

---

**Input:** $\{\mathbf{f}^*\}$, $\{\mathbf{s}_i^{\text{inputs}}\}$, $\{\mathbf{s}_i^{\text{trunk}}\}$, $\{\mathbf{z}_{ij}^{\text{trunk}}\}$, Noise Schedule $[\beta_0, \beta_1, \ldots, \beta_T]$, $\gamma_0 = 0.8$, $\gamma_{\min} = 1.0$, noise scale $\lambda = 1.003$, step scale $\kappa = 1.5$, experimental observation $\mathbf{y}$, guidance scale $\boldsymbol{\eta}$, reference structure $\mathbf{r}$, substructure conditioner flag $b$, substructure indices $I$, batch size $n$, number of atoms $m$

**Output:** $\boldsymbol{\mathcal{X}}_l$          Guided Ensemble

$\boldsymbol{\mathcal{X}}_l \sim \beta_0 \cdot [\mathbf{N}^1, \ldots, \mathbf{N}^n]^T$        $\mathbf{N}^i \sim \mathcal{N}(\mathbf{0}, \mathbf{I}_3), \boldsymbol{\mathcal{X}} \in \mathbb{R}^{n \times m \times 3}$

**for** $\beta_\tau \in [\beta_1, \ldots, \beta_T]$ **do**

    $\boldsymbol{\mathcal{X}}_l \leftarrow \text{CentreRandomAugmentation}(\boldsymbol{\mathcal{X}}_l)$

    $\gamma \leftarrow \gamma_0$ if $\beta_\tau > \gamma_{\min}$ else $0$

    $\hat{t} \leftarrow \dfrac{\beta_{\tau-1}(\gamma+1)}{\sqrt{t^2 - \beta_\tau^2/\beta_{\tau-1}}}$

    $\boldsymbol{\xi}_l \leftarrow \lambda\sqrt{\hat{t}^2 - \beta_\tau^2} \cdot [\mathbf{N}^1, \ldots \mathbf{N}^n]^T$      $\mathbf{N}^i \sim \mathcal{N}(\mathbf{0}, \mathbf{I}_3), \boldsymbol{\xi}_l \in \mathbb{R}^{n \times m \times 3}$

    $\boldsymbol{\mathcal{X}}_l^{\text{noisy}} \leftarrow \boldsymbol{\mathcal{X}}_l + \boldsymbol{\xi}_l$

    $\boldsymbol{\mathcal{X}}_l^{\text{denoised}} \leftarrow \text{DiffusionModule}(\{\boldsymbol{\mathcal{X}}_l^{\text{noisy}}\}, \hat{t}, \{\mathbf{f}^*\}, \{\mathbf{s}_i^{\text{inputs}}\}, \{\mathbf{s}_i^{\text{trunk}}\}, \{\mathbf{z}_{ij}^{\text{trunk}}\})$

    **if** $b$ **then**

        **for** $i \in I$ **do**

            $\boldsymbol{\mathcal{X}}_i^{\text{denoised}} \leftarrow [\mathbf{r}_i, \ldots, \mathbf{r}_i]$         Repeated $n$ times

        **end for**

    **end if**

    $\delta_l \leftarrow (\boldsymbol{\mathcal{X}}_l - \boldsymbol{\mathcal{X}}_l^{\text{denoised}})/\hat{t}$

    $\mathcal{L} \leftarrow \log p(\mathbf{y} | \boldsymbol{\mathcal{X}}, \mathbf{a})$

    $g \leftarrow \dfrac{\partial \mathcal{L}}{\partial \boldsymbol{\mathcal{X}}_l^{\text{noisy}}}$        Guidance Score is with respect to the ensemble

    $g \leftarrow g \cdot \dfrac{\|\delta_l\|_2}{\|g\|_2}$          Gradient Scaling

    $\delta_l \leftarrow \delta_l + g \cdot \boldsymbol{\eta}$

    $dt \leftarrow \beta_\tau - \hat{t}$

    $\boldsymbol{\mathcal{X}}_l \leftarrow \boldsymbol{\mathcal{X}}_l^{\text{noisy}} + \kappa \cdot dt \cdot \delta_l$

**end for**

**return** $\boldsymbol{\mathcal{X}}_l$

---

**Algorithm 2** Selecting samples using matching pursuit (Mallat & Zhang, 1993)

---

**Input:** samples in ensemble $\boldsymbol{\mathcal{X}} = \{\mathbf{X}^1, \ldots, \mathbf{X}^n\}$; experimental observation $\mathbf{y}$; amino acid sequence $\mathbf{a}$; likelihood function to be maximized $\log p(\mathbf{y} | \boldsymbol{\mathcal{X}}, \mathbf{a})$; maximum samples to select $n_{\max}$;

$\mathcal{I} = \emptyset$

$s_{\text{current}} = 0$

**while** $|\mathcal{I}| < n_{\max}$ **do**

    $L = \{\ell_k = \log p(\mathbf{y} | \boldsymbol{\mathcal{X}}_{\mathcal{I} \cup \{k\}}, \mathbf{a}) \; : \; k \in \mathcal{I}^{\text{c}}\}$

    $k^* \leftarrow \arg\max_k L \quad s^* \leftarrow \max_k L$      *Maximize $\log p(\mathbf{y}|\boldsymbol{\mathcal{X}}_{\mathcal{I} \cup \{k^*\}}, \mathbf{a})$*

    $\mathcal{I} \leftarrow \mathcal{I} \cup \{k^*\}$            *Add best sample*

    **if** $s^* < s_{\text{current}}$ **then**

        break

    **end if**

    $s_{\text{current}} \leftarrow s^*$

**end while**

**return** $\boldsymbol{\mathcal{X}}_{\mathcal{I}} = \{\mathbf{X}^k : k \in \mathcal{I}\}$

---

# A. Appendix and Supplemental Material

### A.1. Guidance using AlphaFold3

In this section, we extend Algorithm 18 from the Supplemental Information of AlphaFold3 (Abramson et al., 2024) to incorporate additive guidance using experimental observations. The modified Algorithm is detailed in Algorithm 1.

Below, we describe the hyperparameters and log-likelihood formulations used for density-guidance and NOE-guidance.

For density-guidance, we used equation (1) as the primary log-likelihood function. However, due to the local nature of density-guidance, we apply the density loss function to atoms within a continuous residue region of the amino acid $\mathbf{a}$: $\mathbf{r} = [r_{\min}, r_{\max}]$, where $1 \leq r_{\min} < r_{\max} \leq |\mathbf{a}|$. To ensure stability outside this region, we used the Substructure Conditioner loss in equation (3) to anchor the remaining atoms. Hence, we used the following log-likelihood for density-guidance.

$$\log p(F_{\mathrm{o}} \mid \boldsymbol{\mathcal{X}}, \mathbf{a}) = -\left\| F_{\mathrm{o}} - \frac{1}{n}\sum_{k=1}^{n} F_{\mathrm{c}}(\mathbf{X}^k, \mathbf{a}) \right\|_1 - \frac{\lambda}{n}\sum_{k=1}^{n}\sum_{i\in A} \|\mathbf{x}_i^k - \mathbf{y}_i\|^2$$

Where $A$ is the set of reference atom locations within the residue region $\mathbf{r}$. The specific choice of the $\mathbf{r}$ depends on the protein and is detailed in the latter sections. We used $\lambda = 0.1$ to scale the substructure conditioner. For guidance, we used $\boldsymbol{\eta} = 0.1$ in equation (5).

Unlike density-guidance which focuses *local* fitting, NOE-derived restraints are *global* in nature. Therefore, we do not apply the substructure conditioner when using equation (2) for NOE guidance. For guidance, we evaluated $\boldsymbol{\eta} = 0.3, 0.5$ in equation (5), and selected the parameter based on the number of restrained obeyed.

To ensure numerical stability during guided diffusion, we apply gradient clipping to clip the guidance score. This prevents instability due to large gradients, ensuring a smooth integration of experimental constraints into the diffusion process.

### A.2. Noise model underlying log likelihood

This section discusses the underlying noise models $\mathcal{L}$ that lead to the log-likelihood functions in Equations 1, 2, and 3.

- **Density Loss:** A Laplace noise model is used, where the difference between $F_{\mathrm{o}}$ and $F_{\mathrm{c}}$ is drawn from a Laplace distribution centered at zero with unit scaling. This is given by,

$$\mathcal{L} = F_{\mathrm{o}} - F_{\mathrm{c}} \sim \mathrm{Laplace}(0, 1) \tag{6}$$

  This model, along with the Gaussian model, is used to model electron density. However, a more realistic noise model involves complex physics, as noise is introduced at the level of Fourier intensities. We will address this in future works.

- **NOE Loss:** Instead of a typical noise model, a piecewise function is used to model the underlying noise function.

$$\mathcal{L} = \begin{cases} 0, & \text{if } \underline{d}_{ij} \leq d_{ij}(\boldsymbol{\mathcal{X}}) \leq \overline{d}_{ij} \\ (d_{ij}(\boldsymbol{\mathcal{X}}) - \underline{d}_{ij})^2, & \text{if } d_{ij}(\boldsymbol{\mathcal{X}}) < \underline{d}_{ij} \\ (d_{ij}(\boldsymbol{\mathcal{X}}) - \overline{d}_{ij})^2, & \text{if } d_{ij}(\boldsymbol{\mathcal{X}}) > \overline{d}_{ij} \end{cases} \tag{7}$$

  In Equation 7, if the pairwise distances $d_{ij}(\boldsymbol{\mathcal{X}})$ is between the bounds $\left[\underline{d}_{ij}, \overline{d}_{ij}\right]$, then we assume a uniform noise distribution (0 loss). Otherwise, we assume a Gaussian-like noise distribution (quadratic).

- **Substructure Loss:** A Gaussian distribution with a fixed isotropic covariance is used as the noise model.

$$\mathcal{L} = \mathbf{x}_i^k \sim \mathcal{N}(\mathbf{y}_i, \mathbf{I}) \tag{8}$$

  Here, each atom $\mathbf{x}_i^k$ is drawn from a multivariate Gaussian distribution centered at reference atom location $\mathbf{y}_i$ with identity covariance matrix $\mathbf{I}$ (isotropic).

### A.3. Baselines

We extend the evaluation presented in Tables A3 and A4 for both X-ray crystallography and NMR experimental observations by incorporating additional baselines, specifically AlphaFlow and ESMFlow (Jing et al., 2024). For the X-ray crystallography,

we include a comparison with Chroma ensembles generated using non-i.i.d. guidance (Maddipatla et al., 2024). Table A8 presents a direct comparison between non-i.i.d. guided X-ray crystallography (which consistently outperforms other baselines in Table A3) and the newly added baselines. In all cases, non-i.i.d. guided AlphaFold3 demonstrates superior performance compared to other methods.

Similarly, Table A4 has been extended to include AlphaFlow and ESMFlow in the NMR structure benchmark, as shown in Table A7. Our results indicate that the structural ensembles generated by NOE-guided AlphaFold3 adhere more closely to experimental constraints than those produced by any other baseline. ably, in half of the cases, these ensembles show better agreement with the constraints than the deposited NMR structures resolved using molecular dynamics. It is important to note that due to the absence of explicit sequence conditioning in Chroma, the resulting ensembles diverged significantly from the true structures.

### A.4. Runtime Analysis

In addition to predicting experimentally faithful protein ensembles, the proposed approach renders samples in a computationally efficient manner. As shown in Table A5, an ensemble of 16 proteins with over 300 residues is sampled in approximately 7 minutes using density guidance, with minimal added latency compared to unguided AlphaFold3. Similarly, in Table A6, we generate ensembles of 32 conformations in approximately 14 minutes while modeling distance restraints as ensemble statistics. Our approach is significantly faster than restrained molecular dynamics methods like CYANA (Güntert, 2004), which is the current state-of-the-art technique for NMR structure determination.

### A.5. AlphaFold3 model and hardware resources

Across all experiments in this paper, we used the open-sourced Protenix (Chen et al., 2025) model, a `PyTorch`-based (Paszke et al., 2019) re-implementation of AlphaFold3. However, for the AlphaFold3 baseline comparisons, we report predctions generated using the official AlphaFold3 weights and source code (Abramson et al., 2024). All computations were performed on NVIDIA H100 and L40S GPUs.

### A.6. Calculating $F_c$ from a protein structure

In equation (1), we compute the $L_1$ norm of the difference between $F_o$ and expected value of $F_c$ along the protein ensemble $\mathcal{X}$. Here, $F_c : \mathbb{R}^3 \to \mathbb{R}$ is the calculated electron density determine using the 3D Cartesian coordinates of a specific protein structure $\mathbf{X} = (\mathbf{x}_1, \dots \mathbf{x}_m)$. Formally, $F_c$ can be computed using a sum over a finite number of kernel density estimates,

$$F_c(\boldsymbol{\xi}) = \sum_{q=1}^{N_s} \sum_{i=1}^{m} \sum_{j=1}^{5} a_{ij} \cdot \left( \frac{4\pi}{b_{ij} + B} \right)^{\frac{3}{2}} \cdot \exp\left( -\frac{4\pi^2}{b_{ij} + B} \| \mathbf{R}_q \mathbf{x}_i + \mathbf{t}_q - \boldsymbol{\xi} \|_2^2 \right), \tag{9}$$

where $N_s$ is the number of symmetry operations (Brock et al., 2016), $m$ is the number of atoms in the asymmetric unit, $\mathbf{R}_q$ is the rotation matrix of the $q$-th symmetry operation, $\mathbf{t}_k$ is the translation vector of the $k$-th symmetry operation, $\mathbf{x}_i \in \mathbb{R}^3$ is the location of the $i$-th atom, $a_{ij}$ and $b_{ij}$ are tabulated form factors defined for every heavy atom (Prince, 2004), $B$ is the B-factor, and $\boldsymbol{\xi} \in \mathbb{R}^3$ is the point in Euclidean space where density is calculated. In standard crystallographic pipelines, the B-factor is used to model experimental electron density as a mixture of Gaussians. However, we consider the B-factor to be a bandwidth parameter in kernel density estimation (KDE) techniques (Terrell & Scott, 1992). Ideally, we would like to optimize the B-factor when guiding the diffusion process; however, due to the stochastic nature of the diffusion process, optimizing B-factor proved to be quite unstable and often pushed the diffusion variable off of the diffusion manifold. Hence, we used a uniform B-factor that is inversely related to the size of the ensemble, $B = \frac{4}{n}$, for all the atoms in the ensemble. We consistently performed density guidance on an ensemble of size 16. For some bigger proteins, we used a batch size 12 to avoid out of memory exceptions.

### A.7. Filtering & relaxation after density guidance

Following the guided (non-i.i.d.) diffusion sampling procedure using density maps, we apply a two-stage filtering and refinement procedure to ensure the physical plausibility of the generated structures. To remove structurally invalid samples, we first identify and eliminate structures with broken covalent bonds and/or steric clashes.

To check for structures with broken bonds, we determine all bonded atom pairs within the protein structure using Gemmi (Wojdyr, 2022) and compute their Euclidean distances. A structure is considered to have broken bonds if the distance

between any bonded atoms exceeds $\tau_{\text{bond}} = 2.1$ Å. In addition, to check for structures with steric clashes, we compute all pairwise interatomic distances and classify a structure as exhibiting steric clashes if the distance between any two atoms is less than $\tau_{\text{clash}} = 1.1$ Å.

After the initial filtering, we further refine the remaining samples by relaxing them with AMBER force field (Wang et al., 2004). This process resolves minor bond length deviations and improves structural consistency by ensuring that atomic interactions conform to physically realistic energy landscapes. For this, we use the publicly available ColabFold implementation (Mirdita et al., 2022).

### A.8. Ensemble pruning with matching pursuit

The matching pursuit-based (Mallat & Zhang, 1993) ensemble selection procedure is detailed in Algorithm 2. Below, we describe additional optimizations and hyperparameters used for ensemble selection following density guidance.

Before selecting the ensemble, we optimize a scalar B-factor $B$ to maximize the log-likelihood in equation (1). This step adjusts the bandwidth of the ensemble's theoretical electron density $F_c$ to best fit the observed density $F_o$. While optimizing $B$ during the diffusion process can introduce numerical instabilities, we avoid the issue here because $B$ is optimized after structure generation, filtering, and relaxation is complete. Hence, we do not encounter similar instabilities here. We optimized $B$ using Adam (Diederik, 2015) optimizer with a step size of 1.0 over 100 iterations. The optimized B-factor $B^*$ is used uniformly across all atoms in the remaining structures. Following this, we apply the matching pursuit algorithm to select the best-fit ensemble.

Across all experiments, we set the maximum ensemble size to $n_{\text{max}} = 5$. We heuristically found that the density is well explained by at most 5 samples and adding more samples to the ensemble would overfit the noise in the density map without yielding a considerable increase in cosine similarity. In Figure A6, we plot the normalized cosine similarity against the number of samples in an ensemble of size 16. The cosine similarity plateaus at 5 samples after which we either get a deteriorated fit to the density, or overfit to noise – both undesirable.

For NMR guided ensembles, we do not employ ensemble selection and our results are evaluated on the full ensemble.

### A.9. Electron density pre-processing

Since the electron density maps available in the PDB are mean-centered and lack an absolute scale, we converted them to physical units of density $[e^-/\text{Å}^3]$ using the method described in (Lang et al., 2014). A comprehensive list of proteins used in our experiments and their corresponding residue regions is provided in Tables A1-A3. While most density maps in our dataset are of high resolution, our method performs exceptionally well on density maps with relatively lower resolution like PDB entries 4OLE, 6JF2, and 6QQF.

### A.10. Ensemble bi-modal distribution evaluation

In Section 5, we demonstrated that density-guided diffusion effectively captures structural heterogeneity present in the protein crystal. Specifically, we evaluated this method on altloc regions of proteins listed in Table A3 and observed that our non-i.i.d. density guidance framework consistently outperforms i.i.d. density guidance across most proteins in the dataset. Also, both methods consistently outperform AlphaFold3.

In this section, we describe a quantitative evaluation demonstrating that non-i.i.d. density guidance better captures the bimodality inherent to altloc regions compared to other methods. To this end, we use samples from three sets of experiments, density-guided (i.i.d.), density-guided (non-i.i.d.), and AlphaFold3 (unguided). In each case, we filter, relax (Appendix A.7), and refine the generated ensemble using matching pursuit-based selection (Appendix A.10). To quantify bimodality, we compute the normalized distance of each sample in the ensemble $\mathcal{X}$ relative to the known altloc configurations (A or B). Consider an individual sample $\mathbf{X} \in \mathbb{R}^{m \times 3}$ from the ensemble $\mathcal{X}$ and the reference altloc A and B structures $\mathbf{X}_a, \mathbf{X}_b \in \mathbb{R}^{m \times 3}$, respectively. Following Rosenberg et al. (2024b), we define the signed normalized distance between $\mathbf{X}$

and the reference conformations as

$$d_{\mathrm{a}} = \|\mathbf{X} - \mathbf{X}_{\mathrm{a}}\|_2^2$$
$$d_{\mathrm{b}} = \|\mathbf{X} - \mathbf{X}_{\mathrm{b}}\|_2^2$$
$$\text{Normalized Distance} = \left[1 - \min\left(\frac{d_{\mathrm{b}}}{d_{\mathrm{a}}}, \frac{d_{\mathrm{a}}}{d_{\mathrm{b}}}\right)\right] \cdot \text{sign}(d_{\mathrm{a}} - d_{\mathrm{b}}). \tag{10}$$

This metric provides an interpretable method for quantifying ensemble distributions. Specifically, samples closer to reference conformation A have the normalized distance approaching $-1$, while those closer to reference conformation B have the normalized distance approaching $+1$. Consequently, negative values indicate proximity to mode A, while positive values indicate proximity to mode B. The resulting normalized distance distributions are visualized in Figure A2, A3, and A4. We observe that non-i.i.d. guided sampling with both Chroma and AlphaFold3 achieves bimodal and multimodal distributions behavior with proximity to both reference conformations (positive and negative modes in the plot). Notably, our electron density-guided approach with AlphaFold3 achieves bimodality more consistently than Chroma-based guidance. In contrast, i.i.d. guided sampling with AlphaFold3 achieves a moderate degree of bimodality but is less effective than non-i.i.d. sampling in terms of accurately capturing the full extent of conformational heterogeneity present in the electron density. Furthermore, unguided methods – including AlphaFold3, AlphaFlow, ESMFlow, and Chroma – frequently fail to recover the correct bimodal behavior.

This demonstrates that non-i.i.d. guidance significantly outperforms i.i.d. guidance in modeling bimodal distributions in electron density maps. In addition, both guided approaches perform significantly better than unguided models.

### A.11. Diffusion guidance landscape

In this section, we analyze the loss curves during the density-guided diffusion process, as depicted in Figure A5, over a batch of 16 ensembles of `2ESK` structures (Residue Range: `113-119`). At each diffusion timestep, we log the cosine similarity between $F_{\mathrm{o}}$ and $F_{\mathrm{c}}$ to quantify the alignment between the observed and predicted electron density maps.

We also log the number of bond length violations across all structures in the ensemble, computed using the same methodology described in Appendix A.7. From Figure A5, we observe that without density guidance, the cosine similarity between $F_{\mathrm{o}}$ and $F_{\mathrm{c}}$ remains significantly lower than in the guided case. This demonstrates that incorporating density guidance during diffusion leads to significantly improved density alignment. Furthermore, we identify a critical phase of the diffusion process, between iterations 130 and 175, where the cosine similarity increases the most. This period coincides with a shapr reduction in bond length violations. This suggests that during these iterations, the diffusion model corrects high-frequency structural features. During iterations 100 to 135, we observe significant fluctuations in the confidence bands of the cosine similarity plots. This suggests high variability in bond length violations across different structures in the ensemble, likely indicating that the diffusion model is correcting low-frequency structural features during this phase.

Lastly, after relaxation and filtering, the number of bond length violations reduces to zero, confirming that relaxation effectively resolves structural inconsistencies. However, we note a slight dip in cosine similarity post-relaxation. This could be attributed to overfitting of the guided diffusion process to the electron density maps, leading to subtle structural violations that are later corrected during relaxation.

### A.12. Estimating NOE restraints from an ensemble

**Measuring distance restraints on an ensemble.** Given an ensemble $\boldsymbol{\mathcal{X}}$ and NOE restraints $D = \{(\underline{d}_{ij}, \overline{d}_{ij}) : (i, j) \in \mathcal{P}\}$ of pairs of lower and upper bounds, the NOE restraints are employed on the ensemble average, $d_{ij}(\boldsymbol{\mathcal{X}}) = \frac{1}{n}\sum_{k=1}^{n} d_{ij}(\mathbf{X}^k)$ of the distances $d_{ij}(\mathbf{X}^k) = \|\mathbf{x}_i^k - \mathbf{x}_j^k\|$ between pairs of atoms $i, j$ in individual structures $\mathbf{X}^k$, as written in equation (2).

**Heavy atom approximation.** AlphaFold3 models only heavy atoms (i.e., it does not include hydrogen atoms), whereas all NOE restraints—based on internuclear interactions—are defined between hydrogen atoms. To address this discrepancy *during guidance*, each NOE restraint in experimental data $D$, initially specified between hydrogen atoms, is approximated by applying an equivalent distance constraint to their covalently bonded heavy atoms (e.g., N for NH or C for CH groups). Given fixed bond lengths (N–H: 1.0Å, C–H: 1.1Å), the maximum error introduced by this substitution is $4.4\text{Å} = 2 \times (1.1\text{Å} + 1.1\text{Å})$. *During evaluation*, hydrogen atoms are placed into the model at the relaxation stage (Section 4.2), enabling *correct evaluation* of NOE restraints against explicit hydrogen positions without employing the heavy atom approximation.

**Distances vs. peak intensities** NOE measurements in NMR arises from dipolar interactions between nuclei (typically protons) within $\sim 6\text{Å}$. The intensity of NOE cross-peaks in a NOESY spectrum is inversely proportional to the sixth power of the interatomic distance $\left(I \propto 1/r^6\right)$. Post peak assignment, the cross-peak intensities are converted into distance restraints via $r = r_{\text{ref}} \left(I_{\text{ref}}/I\right)^{1/6}$, where $r_{\text{ref}}$ and $I_{\text{ref}}$ are the distance and peak intensity of a reference nuclei pair. To ensure physical accuracy, the NOE-implied distance between two atoms in an ensemble should be computed by first averaging the calibrated peak intensities, and then converting the resulting mean intensity back into a distance. We will adopt this more rigorous approach in our follow-up work.

## A.13. NMR data collection and pre-processing

NOE-based distance restraints were extracted from NMR STAR files corresponding to the PDB entries. The `pynmrstar` library was employed to parse these restraints, with explicit selection of NOE-derived constraints (categorized under `_Gen_dist_constraint_list.Constraint_type`). All non-NOE structural restraints (e.g., hydrogen bonds, dihedral angles, or RDC-derived constraints) were excluded to focus specifically on distance geometry derived from NOEs. Ambiguous NOE assignments involving multiple proton pairs were retained without filtering to reflect inherent NMR uncertainty and are considered for evaluation. Distance lower and upper bounds were directly obtained from the STAR file fields `_Gen_dist_constraint_list.Distance_lower_bound_val` and `_Gen_dist_constraint_list.Distance_upper_bound_val`, respectively. Missing lower bounds were explicitly set to 0 Å to enforce a physically meaningful minimum distance. Parsed bounds were retained in Ångström units as provided, with no additional thresholding or normalization applied to preserve the experimental restraint set. These bounds were used directly as inputs to the likelihood term (Section 3.2), which models NOE-derived distance uncertainties through a truncated super-Gaussian potential acting between the parsed lower and upper bounds.

## A.14. Computational estimation of N-H bond order

The N-H bond order parameter $S^2$ captures the backbone flexibility. Physically, $S^2$ measures the long-time limit of the autocorrelation function of the N-H bond vector. Given a structural ensemble $\boldsymbol{\mathcal{X}}$, the N-H bond order can be computationally *estimated* as follows (Palmer, 2004):

1. Align the ensemble to a reference structure $\mathbf{X}_{\text{ref}}$, to account for rotational and translational symmetries.

2. For every structure $\mathbf{X}_k \in \boldsymbol{\mathcal{X}}$ in the ensemble, compute the normalized N-H bond-vectors $\{\mathbf{d}_i^k\}$ at all residues $i$.

3. Given the normalized N-H bond-vector, the calculated bond-order $S_{\text{c}}^2$ is given by

$$S_{\text{c}}^2 \approx \frac{1}{n(n-1)} \sum_{k \neq l}^{n} P_2(\mathbf{d}_k \cdot \mathbf{d}_l),$$

where $n = |\boldsymbol{\mathcal{X}}|$ is the ensemble size, and $P_2(x) = \frac{1}{2}\left(3x^2 - 1\right)$ is the Legendre polynomial of order 2.

## A.15. Evaluation metrics used in NMR structure determination experiments

In what follows, we describe how we compute quantitative metrics reported in Table A4 and Figure A7. Given a structural ensemble $\boldsymbol{\mathcal{X}}$ and a restraint list $D = \{(\underline{d}_{ij}, \overline{d}_{ij}) : (i,j) \in \mathcal{P}\}$, we measure the quality of $\boldsymbol{\mathcal{X}}$ in terms of its adherence to $D$ and as a secondary measure, we evaluate the recovered conformational flexibility with respect to the NMR ensemble $\boldsymbol{\mathcal{X}}_{\text{NMR}}$. To evaluate restraint violation across the ensemble, we first compute the ensemble distance matrix $D(\boldsymbol{\mathcal{X}})$, whose entries are given by $d_{ij}(\boldsymbol{\mathcal{X}}) = \frac{1}{n} \sum_{k=1}^{n} d_{ij}(\mathbf{X}^k)$.

**Restraint violation percentage (Viol. %)** For each experimental restraint in $D$, a violation occurs if the ensemble averaged distance $d_{ij}(\mathcal{X})$ lies outside the interval $[\underline{d}_{ij}, \overline{d}_{ij}]$. Viol. % is the percentage of restraints in $D$ that are violated.

**Restraint violation distance (Viol. Å).** For each restraint, the violation magnitude is the absolute deviation of from the nearest bound (lower or upper) if outside the interval; otherwise, it is zero. Viol. Å is the average of these deviations across all restraints in $D$.

**Handling ambiguous restraint groups.** In NMR experiments, distance restraints are often organized into `restraint groups` $\mathcal{G} \subseteq D$ with intra-group `OR` conditions to account for assignment ambiguity. For violation metrics, we compute the effective violation of a group $\mathcal{G}$ as:

$$\text{Viol}(\mathcal{G}) = \min \big\{ \text{Viol}(g) \mid g \in \mathcal{G} \big\}$$

where $\text{Viol}(g)$ is the violation of constituent restraint $g$ (measured via Viol.% or Viol. Å). This ensures a group is considered satisfied if at least one constituent restraint complies with the experimental bounds.

**$\rho$-RMSF with respect to $\mathcal{X}_{\text{NMR}}$.** Given $\mathcal{X}$, this metric measures the correlation in conformational flexibility with respect to $\mathcal{X}_{\text{NMR}}$. We first align $\mathcal{X}$ and $\mathcal{X}_{\text{NMR}}$ to a reference structure $\mathbf{X}_{\text{ref}}$ to account for rotational and translational symmetries. Then, for each residue location $i$ in the ensemble $\mathcal{X}$, the root mean square fluctuation (RMSF) is computed as,

$$\text{RMSF}^i_{\mathcal{X}} = \sqrt{\frac{1}{n} \sum_{k=1}^{n} \big\| \mathbf{x}_i^k - \overline{\mathbf{x}}_i \big\|^2}, \quad \text{where } \overline{\mathbf{x}}_i = \frac{1}{n} \sum_{k=1}^{n} \mathbf{x}_i^k,$$

and $\mathbf{x}_i^k$ represent the coordinates of the CA atom in residue $i$ in sample $k$ of the ensemble. The $\rho$-RMSF is the Pearson correlation coefficient between the RMSF profiles of $\mathcal{X}$ and $\mathcal{X}_{\text{NMR}}$. This effectively measures their similarity in residue-wise conformational flexibility. Higher $\rho$-RMSF values indicate better agreement in flexibility patterns.

