# OpenReview forum: "Inverse problems with experiment-guided AlphaFold"
_ICML.cc/2025/Conference — ICML 2025 poster_

### Official Review · Reviewer_LiAh · 2025-03-11

**Overall Recommendation:** 4

**Summary:**

In this paper, the author introduce a method to guide diffusion-based structure prediction models (e.g., AlphaFold 3) with experimental data to sample conformational ensembles.

**Claims And Evidence:**

The claims are well supported by the results shown in the paper.

**Essential References Not Discussed:**

However, the authors do not cite important prior work in this space (i.e., guiding a protein diffusion model with experimental data). The following references should be discussed:

Fadini, Alisia, et al. "AlphaFold as a Prior: Experimental Structure Determination Conditioned on a Pretrained Neural Network." bioRxiv (2025).
Liu, Yikai, et al. "ExEnDiff: An Experiment-guided Diffusion model for protein conformational Ensemble generation." bioRxiv (2024): 2024-10.
Maddipatla, Sai Advaith, et al. "Generative modeling of protein ensembles guided by crystallographic electron densities." arXiv preprint arXiv:2412.13223 (2024).
Levy, Axel, et al. "Solving inverse problems in protein space using diffusion-based priors." arXiv preprint arXiv:2406.04239 (2024).

**Experimental Designs Or Analyses:**

The experimental setup for testing the possibility of guiding AlphaFold 3 with static electron density maps is valid.

However, in these experiments, most of the structure is also guided using the Substructure Conditioner loss (eq. 3). This is not, per se, an issue but:
* I only saw this information in the supplementary material and believe this would important emphasize this point in the main text to avoid confusion regarding what the method is capable of.
* The authors should discuss:
  * Where does y (the Cartesian coordinates of the atoms in the anchored region) come from?
  * How should we choose the set of anchored atoms (A)? It seems that we need to know in advance the regions where "new" conformations are likely to appear.
  * What is the influence of the size (in terms of consecutive residues) of the non-anchored region? Does the method perform significantly worse when long regions are not anchored?

Furthermore, the maximum number of structure in the ensemble is set to 5.
* Does that limit the applicability of the method to cases where the conformational ensemble only has a few modes?
* Due to what constraints was this maximum number chosen?

**Methods And Evaluation Criteria:**

The baseline (AlphaFold 3) and the test sets seem relevant to evaluate the method.

**Other Comments Or Suggestions:**

Typo on L139 (right column): [I]n some cases...

**Other Strengths And Weaknesses:**

The description of the forward models is clear and accurate. The method is well described and seems reproducible.

**Questions For Authors:**

Please see my questions in "Experimental Designs Or Analyses".

**Relation To Broader Scientific Literature:**

To the best of my knowledge, this is the first method that applies the idea of guiding a pretrained diffusion model with *experimental* data to sample *conformational ensembles*.

**Theoretical Claims:**

The paper does not introduce new theoretical claims.

In equations (1), (2) and (3), the authors should specify the underlying noise model, leading to the given log-likelihood functions.

---

> ### Author Rebuttal · Authors · 2025-04-01
>
> We thank the reviewer for their review.
>
> **Underlying noise model.**
>
> *Eq. 1.* We assume a Laplace noise model. Here, the difference between $F_{o}$ and $F_{c}$ is drawn from a Laplace distribution centered at zero with unit scaling. This model, along with the Gaussian model, is used in electron density modelling. However, a more realistic noise model involves complex physics, as noise is introduced at the level of Fourier intensities. We will address this in future works.
>
> *Eq. 2*: Instead of a typical noise model, we assume a piecewise noise model. If the distance is between the lower and upper bounds, then we assume a uniform noise distribution (0 loss). Otherwise, we assume a Gaussian-like noise distribution (quadratic loss). This model is common in NMR structure modeling (Lindorff-Larsen et al., 2005).
>
> *Eq. 3*: We assume a Gaussian distribution with a fixed isotropic covariance as the noise model. Each $\mathbf{x}\_{i}^{k}$ is assumed to be drawn from $\mathcal{N} (\mathbf{y}_{i}, \mathbf{I})$
>
> **Substructure Conditioner**
>
> Due to space constraints, we could not include the substructure conditioner in the main text. We will move it there in the future. Re the other questions:
> - $y$ is from AF3 predictions.
> - To select the set of anchored atoms, we examine AlphaFold3 (AF3) predicted structures alongside density maps to identify regions that are either not faithful to the map or exhibit structural heterogeneity in the map. We then extract a 3D slice of the map around the region, including all the relevant atoms and applying a padding of 5Å along each axis.
> - No, the method does not yield significantly worse results when long regions are not anchored. For instance, in Figure 2A and 3 (of manuscript), a 10-residue region is not anchored by the conditioner. Yet, we recover conformers that fit the density map well and the cosine similarity is comparable to PDB structures. We conducted additional experiments on 15 structures with conformational heterogeneity ([link](https://postimg.cc/XZt5h686)). For some, we optimized regions spanning up to 22 residues (5v2m and 6e2s), and for all cases, the cosine similarity is on par with the PDB (or better). We also included additional baselines like AlphaFlow and ESMFlow (Jing et al., 2024). However, it must be noted that the residue range length does increase the runtime for fitting an ensemble to our experimental observation ([link](https://postimg.cc/qgJK5wcR)).
>
> **Maximum Samples in Ensemble**
> - No, this method is effective even when the sample has a single mode too. For instance, all structures in Tabs. A1 & A2 (in manuscript) have a single mode. Here, the ensemble (selected using algorithm 2) consists of similar looking structures without the separation we typically observe in structures with multiple conformations. Additionally, the ensemble in Figs. 2 & A1 (single mode structures) were selected using Algorithm 2 (manuscript).
> - We heuristically found that the density is well explained by at most 5 samples and adding more samples to the ensemble would overfit the noise in the density map without yielding a considerable increase in cosine similarity. In the attached Figure ([link](https://postimg.cc/BP2nP0WB)), we plot the normalized cosine similarity against the number of samples in the ensemble of size 15. We see the cosine similarity stabilizes at 5 samples. Post that, we either get a deteriorated fit to the density, or overfit to noise – both undesirable.
>
> **Missing literature review**
> - Fadini et al.: This work was released after the ICML deadline. This work fits a single protein conformer to the static electron density map by optimizing AlphaFold2’s MSA contact maps. However, they are unable to account for conformational ensembles (15% of proteins exhibit non-trivial conformational heterogeneity in the same protein). Proteins are better described as ensembles rather than single conformers.
> - Liu et al.: This study samples structural ensembles using the str2str protein diffusion model, with guidance from cryo-EM experimental observation. While related, their focus is on a different modality than ours and does not address crystallographic density fitting or NOE observations.
> - Maddipatla et al.: This method fits ensembles to crystallographic density maps using Chroma and captures multiple conformers to some extent. We observed that it fails to capture conformational heterogeneity when optimizing over a long residue range due to Chroma’s hierarchical formulation. Moreover, because the sequence conditioning in Chroma is only “promoted” and not imposed, we found it impossible to impose “global” structural constraints as those in the case of NMR.
> - Levy et al.: They also use Chroma as their protein diffusion model and inherit similar problems with packing sidechains. Additionally, they do not use experimental electron density maps or distance restraints, but instead rely on synthetic data.
>
>
> We will update the manuscript to include all aforementioned points.

---

### Official Review · Reviewer_kpSM · 2025-03-14

**Overall Recommendation:** 3

**Summary:**

This paper introduces Experiment-Guided AlphaFold3, a framework that integrates experimental data with deep learning priors to generate structural ensembles aligned with experimental observables. Standard protein structure predictors like AlphaFold3 produce single static structures, failing to capture conformational heterogeneity. The proposed method adapts AlphaFold3’s diffusion-based sampling to incorporate experimental constraints, refines structures via force-field relaxation, and selects ensembles maximizing agreement with experimental data. Density-Guided AlphaFold3 improves crystallographic modeling by generating structures more faithful to electron density maps. NOE-Guided AlphaFold3 refines ensembles to satisfy NMR-derived distance restraints, better capturing protein dynamics. The approach significantly reduces computational time compared to traditional crystallographic and NMR workflows. Results show enhanced structural accuracy over standard AlphaFold3 and, in some cases, even PDB-deposited structures. This work advances protein structure modeling by bridging the gap between deep learning predictions and experimental measurements.

**Claims And Evidence:**

The paper primarily validates its approach through specific case studies, such as ubiquitin for NMR and selected crystal structures for X-ray modeling. However, it remains unclear how well the method generalizes across diverse protein families, particularly for flexible or disordered proteins and multi-domain assemblies. Without broader benchmarking, the extent to which Experiment-Guided AlphaFold3 captures conformational heterogeneity across structurally varied proteins is uncertain. Additional validation on a wider range of experimental datasets would strengthen the claim of broad applicability.

**Essential References Not Discussed:**

No.

**Experimental Designs Or Analyses:**

The experimental design is reasonable but has limitations in scope and validation. The method is tested on a few case studies (e.g., ubiquitin, selected crystal structures), raising concerns about generalizability to diverse protein families. The claim of computational efficiency is not backed by explicit runtime comparisons. Additionally, statistical analyses (e.g., RMSD distributions, significance testing) are missing. Broader benchmarking and more rigorous validation would strengthen the conclusions.

**Methods And Evaluation Criteria:**

The proposed methods and evaluation criteria are reasonable and relevant, but they currently rely on a limited set of case studies. A more diverse benchmarking strategy, additional validation metrics, and explicit runtime comparisons would improve the strength of the evaluation.

**Other Comments Or Suggestions:**

I initially feel inappropriate to accept this paper due to the weak experiments and the uncommon writing style. I am not a biologist, so I will consider raise my initial score by carefully reading other reviewers' opinions and seeing my major concerns are well addressed.

## Update after rebuttal
I appreciate the authors' significant efforts in conducting additional experiments, which have strengthened the paper. As a result, I have decided to raise my score. However, the unconventional writing style—unusual for a machine learning conference—still prevents me from assigning a higher score, despite finding the paper more interesting with the new results.

**Other Strengths And Weaknesses:**

A key concern is that the paper's primary audience seems to be structural biologists rather than the machine learning (ML) community. The writing style and structure are uncommon for an ML paper, with extensive discussions on case studies and biological validation rather than a focus on methodological advancements or generalizable ML insights. While this depth may make it a high-quality protein design or structural biology application paper, it is unclear whether it aligns with ICML’s core focus on ML innovation. The paper would be more suitable for ICML if it placed greater emphasis on the ML contributions, generalizable techniques, and broader computational impact rather than domain-specific experimental results.

**Questions For Authors:**

No more questions.

**Relation To Broader Scientific Literature:**

This paper builds on recent advances in deep learning-based protein structure prediction, particularly AlphaFold3 (Abramson et al., 2024), and addresses its limitation in capturing conformational heterogeneity. While previous methods like Rosetta (Baek et al., 2021) and MD-based NMR refinement (Lindorff-Larsen et al., 2005) have incorporated experimental data, they are computationally expensive. The work also relates to AlphaFlow (Jing et al., 2024) and ensemble modeling approaches in X-ray crystallography (Furnham et al., 2006; van den Bedem & Fraser, 2015) and NMR. However, it uniquely leverages AlphaFold3 as a structure prior, integrating experimental constraints to generate physically realistic and experimentally consistent structural ensembles. By bridging deep learning-based structure prediction with experimental refinement, this work contributes to both computational and experimental structural biology.

**Theoretical Claims:**

There are no theoretical claims in this paper.

---

> ### Author Rebuttal · Authors · 2025-04-01
>
> We thank the reviewer for their comments.
> > tested on a few case studies (e.g., ubiquitin, selected crystal structures)
>
> This paper is as a proof of concept that AlphaFold3 (AF3) can be guided by experimental observations to generate heterogeneous ensembles.
>
> We note that there was extensive quantitative evaluation presented in the Appendix (crystallography: Tab. A1-3 – 44 structures; NMR: Tab. A4 – 13 structures in the manuscript). Our case studies were curated in order to showcase the method in biologically and methodically interesting settings. For e.g., Ubiquitin is the benchmark protein for NMR structure determination, and was the sole subject of study of highly influential works in Nature (Lindorff-Larsen et al., 2005) and Science (Lange et al., 2008). It is essential for any NMR structure determination work to benchmark their performance on Ubiquitin.
>
> In crystallography, our benchmarks comprise several “difficult” and interesting cases that we curated, which AF3 consistently fails on (Tabs. A1 – 3). As control, we cover “simple” cases, where AF3 performs well, to show our approach does not deteriorate AF3’s performance (Tab. A2).
>
> That said, considering the reviewer’s request, we curated an additional set of results for both crystallography (+15 proteins) and NMR (+27 proteins). The updated quantitative results are available at these links: https://postimg.cc/XZt5h686, https://postimg.cc/Wtws9D06.
>
> We added new baselines to existing tables (Tab. A3 and A4): https://postimg.cc/6yhdQS9C, https://postimg.cc/FYQKksH6
>
> We are currently extending this study to refit the entire PDB.
> > Unclear how well the method generalizes across diverse protein families
>
> - *Flexible proteins:* AF3 structures are generally rigid and do not accurately capture the flexibility of proteins. Protein flexibility is captured by NMR in two ways: (i) solution-state NMR, the proteins are generally more flexible in solution compared to crystals, (ii) NMR order parameters, e.g. N-H S², measure the backbone flexibility of a protein because they capture the time-expectation of the N-H bond vector. In Fig. A3, we compare the backbone flexibility of the recovered ensemble against the experimentally measured order parameters. We also demonstrate that we successfully capture this flexibility, but unguided AF3 does not.
>
> - *Disordered proteins:* Because AF3 was trained on crystal structures, it fails to predict intrinsically disordered proteins. While very interesting, this is beyond the scope of this work and deserves a dedicated study.
>
> - *Multidomain assemblies:*  One of the reasons we chose AF3 as the generative model is its ability to predict the structure of protein-ligand and multi-protein complexes. We currently have promising preliminary results of extending the proposed methods to such cases, however, we believe that this deserves a dedicated study.
>
> > primary audience seems to be structural biologists than ML community
>
> We understand the reviewer’s concern. We agree that our approach is of immediate use to structural biologists, and our writing style might be unorthodox to the ICML audience. Yet, it was the ML community that pioneered the development of protein structure generative models, such as AlphaFlow (ICML 2024). Our perspective is that one of the primary uses of a protein structure generative model is its usage as a structural prior while solving inverse problems. To our knowledge, ours is the first work that employed these structural priors to solve real-world inverse problems. This is inherently a computational problem (not a biological one) addressed with computational tools. While we believe that the tools developed in our work directly impact structural biology workflows, they also introduce a new benchmark for validating future developments in modeling protein structure priors on different downstream tasks. Furthermore, our work might inspire different ways of solving these important real-world inverse problems.
> > computational efficiency, runtime comparisons
>
> For crystallography (Table: https://postimg.cc/qgJK5wcR), our approach samples a batch of 16 proteins in ~7 minutes for 300+ residue systems, adding minimal latency vs. unguided AF3. For NMR (Table: https://postimg.cc/jndm3jrZ), we generate ensembles of 20 conformations in ~10 minutes—significantly faster than restrained MD methods like [CYANA](https://link.springer.com/article/10.1007/s10858-015-9924-9), while modeling distance restraints as true ensemble statistics.
> >Broader benchmarking and more rigorous validation would strengthen the conclusions.
>
> To validate NMR, we evaluate backbone flexibility using N-H S² parameters (Fig. A2). For crystallography, we quantify ensemble heterogeneity via bimodality scores (subsection A3/Fig. A2), outperforming baselines in comparative benchmarks. (https://postimg.cc/F794Vc04, https://postimg.cc/dh5rV8jt, https://postimg.cc/VS5MrdK9). We note that we _don't_ guide our ensembles on either of the validation metrics.

---

### Official Review · Reviewer_GYEr · 2025-03-16

**Overall Recommendation:** 3

**Summary:**

This paper proposes to use AlphaFold3 as a structure prior for protein crystal structure determination from cryo-EM or NMR experiments. Specifically, it uses AlphaFold3 to predict the initial structure and then use the gradient from the electron densities to guide the diffusion module of AlphaFold3 to refine its structure to agree with experimental observation. The method is evaluated on crystallographic electron density data in PDB and shows good performance.

**Claims And Evidence:**

The claims made in this paper looks convincing.

**Essential References Not Discussed:**

[1] This paper should cite Accelerating crystal structure determination with iterative AlphaFold prediction, Structure Biology 2023

**Experimental Designs Or Analyses:**

The experimental design would benefit from more baselines, such as [1].

[1] Accelerating crystal structure determination with iterative AlphaFold prediction, Acta Crystallogr D Struct Biol, 2023

**Methods And Evaluation Criteria:**

The evaluation criteria make sense, but the experiment section would benefit from more baseline.

**Other Comments Or Suggestions:**

Suggestion: the author should move some important figures to the main paper rather than in the supplementary material.

**Other Strengths And Weaknesses:**

The proposed method is the first method that uses classifier guidance to refine AlphaFold3 predicted structure. This contribution is important.

**Questions For Authors:**

N/A

**Relation To Broader Scientific Literature:**

This paper is the first method that uses classifier guidance to refine AlphaFold3 predicted structure. This contribution is original.

**Theoretical Claims:**

There is no theoretical claims in this paper

---

> ### Author Rebuttal · Authors · 2025-04-01
>
> We thank the reviewer for their review.
>
> **More baselines**
>
> Following the reviewer’s suggestion, we extended the evaluation for both the X-ray crystallography and the NMR experimental observations.
> We include the following additional baselines:
> - AlphaFlow (Jing et al., 2024)
> - ESMFlow (Jing et al., 2024)
>
> And specifically for X-ray crystallography altlocs benchmark (Table A3) we also compared against:
> - Chroma (Ingraham et al., 2022)
> - Chroma-guided (Maddipatla et al., 2024)
>
> The updated tables for X-ray have been included in this [link](https://postimg.cc/6yhdQS9C).
>
> **X-ray Crystallography.**  We particularly focused on Table A3 as it captures structural heterogeneity (altlocs). We note that, in all cases, guided AlphaFold3 (AF3) outperforms all other baselines, and in some cases, we are able to fit structures to density better than the structures deposited to the PDB.
>
> We further extended the violin plot in Figure A2 to include the aforementioned baselines ([link1](https://postimg.cc/F794Vc04), [link2](https://postimg.cc/dh5rV8jt), [link3](https://postimg.cc/VS5MrdK9)). We notice that our method captures bimodality better than all other methods (including recent experiment-guided approaches from Maddipatla et al. 2024). We hope that these baselines help justify our claims even further.
>
> *Regarding Terwillinger et al. 2023 [1]*: We attempted to include the comparison to [1] as the reviewer suggested, however, we did not find a publicly available codebase for [1]. We hope that the reviewer can understand that it is difficult to replicate a comprehensive study in this timeframe. Additionally, based on our understanding of the paper, the goal of this paper was not to recover structural heterogeneity given the density.
>
> **NMR.** In a similar vein, we added comparison to AlphaFlow and ESMFlow (Jing et al. 2024) for the NMR structure determination benchmark (Table A4).  We attempted running Chroma (both unconditional and NOE-conditioned), however, we observed that due to lack of explicit sequence conditioning in Chroma, the produced ensembles completely deviated from the true structures. Our results suggest that the structural ensembles produced by NOE-guided AlphaFold adhere to the constraints better than all other baselines, and in half of the cases, they produce a better agreement to the constraints compared to the deposited NMR structures resolved with MD, while taking a tiny fraction of the runtime ([link](https://postimg.cc/FYQKksH6)).
>
> **Note Regarding “Relation to Broader Scientific Literature:”**
>
> While we are the first to use classifier guidance to refine AF3 predictions, we would like to emphasize that we are proposing a new approach to perform classifier guidance on diffusion models. Specifically, regular classifier guided diffusion is i.i.d. in that it encourages each sample in the batch to independently reduce a loss function. However, we are proposing a non-i.i.id. classifier guided diffusion that encourages the entire batch to reduce a loss function.
>
> **Not citing Terwillinger et al. 2023 [1]**
>
> We apologize to the reviewer for oversight. We will add a comprehensive discussion to [1], along with references suggested by Reviewer 4.
>
> We hope this clears all concerns.

---

### Official Review · Reviewer_v7F4 · 2025-03-18

**Overall Recommendation:** 3

**Summary:**

This paper examines the inverse problem of resolving protein structure from experimental data and capturing the heterogeneity arising from the dynamic nature of proteins as an ensemble. To do so, it guides the diffusion module of AF3 with experimental data to satisfy NOE constraints and substructure likelihoods.

**Claims And Evidence:**

I think the key claim here is that we can get a better ensemble of structures out of AF3 that obeys structural heterogeneity if we already have our hands on some x-ray crystollography or NMR data, or known atomic substructures. On the case study proteins, shown in the Appendix, the experimental data guided results have better cosine similarity. The authors also introduce an algorithm to select samples based on the matching pursuit algorithm.

**Essential References Not Discussed:**

not qualified to comment, but baselines are generally too vague / doesn't seem to match the problem statement in Section 3

**Experimental Designs Or Analyses:**

Three data terms are considered:
* crystallographic electron density maps
* nuclear Overhauser effect (NOE) restraints
* sub-structure conditioning using known atom locations

**Question**: what is used as ground truth here? Do you still solve the inverse problem in some other way in order to assess the solution from the AF3-guided method?

From the way I understand it right now, the experiment is almost saying "if you have the result, and guide the prediction with the result, you'll preform better", which feels trivial -- I'll be coming back to this point throughout the review, but I think it's hard for me to appreciate the rest of this paper without clearing up this confusion.

**Methods And Evaluation Criteria:**

**Method**: The method "hacks" the diffusion module in AF3, which outputs an ensemble of structures. The method "refines" this output based on given experimental data. The likelihood of the experimental observation is calculated given each individual ensemble member (for substructures) and for the ensemble average (for electron density and interatomic distances). At the end, AMBER relaxation is used, similar to AF2. Most of the results are in the Appendix. Quantitative metrics include cosine similarity. Results are calculated on selected case study proteins.

**Evaluation**: The evaluation criteria is frankly pretty unclear to me -- as I'll describe throughout this review, I'm not sure why the baseline of AF3 is chosen since it accepts such different inputs (ie. it seems trivial to compare a version of the structure prediction model _with_ the structural experiment data as input and a version _without_ it).

**Other Comments Or Suggestions:**

Nitpicks:
* using the world "inverse problems" in the title feels too broad; can we use something more specific here?
* There isn't really a "results" section in the main text. We have a description of the experiment, and the results described in words all in the same paragraph, and need to flip to the appendix to find the results. From a subjective presentation perspective, I'd prefer to see a smaller / less developed Figure 2 and see more of the graphs in the original text.
* Related works are sprinkled throughout the paper; this makes it hard for the ML audience to directly understand what the comparable problems are.
* The article makes frequent allusions to NOE in the introduction without explaining what it is except the abbrv. -- this is a simple fix that can really make this work more accessible to more parts of the ML audience.

Overall: this seems like a really interesting paper! With the current presentation, though, it seems more useful at a journal where it can reach the right domain experts

**Other Strengths And Weaknesses:**

Strengths:
* As I'll mention throughout this review, I don't fundamentally understand why we need to predict the structure if we already have structure experimental results, though I think this could just be due to me not being familiar enough with this subject area. I think guiding AF3 is a pretty cool engineering feat, but I don't understand this problem enough to comment on how this is better than previous methods.

Weaknesses:
* As I understand it right now, this problem is sort of a trivial one from the ML perspective (i.e. if you guide the prediction with the result, then of course it should do better than not having the result, right?) -- I'm hoping maybe this is only because I'm missing some key piece of understanding, so I'm happy to have a conversation about this during the discussion period.
* Related to my overall confusion about the problem setting: the presentation of the paper is not very well-suited for a ML conference. For example, there aren't any baselines except what is already deposited in the PDB, and it's pretty hard to find what's used for ground truth for the evaluations in Section 6

**Questions For Authors:**

I've sprinkled forms of these questions through out the review, but for completeness:
1. Why we can't just get the structure from the experimental NMR /x-ray crystallography data using the existing ways of solving this inverse problem, rather than go through AF3-guidance?
2. Why were these baselines (AF3, existing PDB deposits) chosen? These seem to be solving a very different problem setting, with very different input information available.
3. What is the ground truth that's used for the results in Section 6?
4. Am I correct in understanding that the problem setting we're looking at is that of resolving the original heterogeneous protein structure from the experimental NMR/x-ray crystallography data? If this is the case, it feels like that we should be comparing to CryoDRGN etc.

Again -- I don't feel fully confident in my understanding of this area, so I'm happy to have a discussion about this with the authors.

**Relation To Broader Scientific Literature:**

not qualified to comment, but my (possibly uninformed) hunch is that actual methods for solving structure from NMR/etc. is more useful, like CryoDRGN etc.?

**Theoretical Claims:**

n/a

---

> ### Author Rebuttal · Authors · 2025-04-01
>
> We hope our answers explain the purpose and novelty of the proposed framework and invite the reviewer to ask further questions.
>
> **Clarification 1: The problem is not trivial**
>
> Raw experimental observations do not contain the atomic model. **In crystallography**, the raw experimental data are intensities of the X-ray diffraction patterns collected at different orientations. Phases, required to reconstruct the electron density, cannot be directly measured and are estimated to generate a rough electron density map that is iteratively refined (partially manually) by improving the fit of the model to the density map that is improved by better phase estimates from the improved model.
>
> Traditionally, the structure is represented as a product of marginal distributions, i.e., Gaussian distributions centered at atom locations with B-factors representing the uncertainty within the atom position. This representation is limited since: 1. ~15% of protein structures exhibit multiple backbone conformers (modes) reinforcing the fact the electron density is an ensemble measurement, 2. It loses information crucial for mechanistic structural biological studies.
>
> We claim to be probably the first approach to sample from the joint distribution of the entire set of atoms. This is possible because of AF3 sequence-conditioned diffusion model. We believe that ensembles of realizations sampled from the joint distribution are necessary to study complicated effects like allostery. We are currently working on refitting the structures on the PDB with ensembles instead of B-factors.
>
> In **biomolecular NMR**, through-space dipolar couplings are measured and assigned to imply pairwise distance restraints within the structure/ensemble. Recovering the structure/ensemble from distance restraints requires restrained molecular dynamics simulations, taking hours to days to produce one sample. Typically hundreds of proteins are simulated, and the ~20 lowest energy ones are selected. Not only is this approach time-consuming, it also fails to treat the NOE restraint as an ensemble measurement [Fowler NJ et al., 2022 Structure]. Attempts at ensemble MD [Lindorff-Larsen et al., 2005,  Lange et al., 2008], are extremely computationally expensive and have been tested on only a handful of proteins. Our approach accelerates this workflow by at least two orders of magnitude. We are currently running NMR-based structure determination for the entire PDB. We believe downstream analyses of the refitted NMR structures could potentially lead to scientific discoveries.
>
> **Major concern 2: Baselines.**
>
> Our method is not an “improved version of AlphaFold” and solves a different problem. AF3 is an inductive model capable of predicting the structure from the sequence in the absence of any experimental input. We use AF3 as a prior in solving the inverse problem of recovering the structure given the experimental input (transductive method). AF3 performs well on many protein regions and our benchmarks were chosen as “difficult” and interesting cases where AF3 fails consistently. We show that guiding with experimental data produces structures consistent with the experiment and often fits the electron density better than the PDB structures.
>
> We compare the PDB structure as the state of the art solution and unconditional AF3, to quantify the information gained due to the conditioning by the experimental input. “Simple” cases in which AF3 already performs well, emphasize that the experimental guidance does not deteriorate performance (Tables in the Appendix). We extended our experiments to include additional baselines and proteins. The updated quantitative results are available at the following link - X-ray: https://postimg.cc/XZt5h686, NMR: https://postimg.cc/Wtws9D06.
>
> *No real “groundtruth” exists*. The observables contain only indirect information about the underlying atomic structural ensemble and cannot be compared to directly. The electron density does not contain atom labels, and in NMR only distance constraints are observed directly. Hence there is no real “ground truth”. However, we still quantify and report quality of fit criteria as customary in the field.
>
> **Minor comments:**
> > CryoDRGN
>
> CryoDRGN solves inverse problems in cryoEM, outputting electrostatic potential maps, not atomic models and is not relevant to the proposed methods.
> > NOE
>
> Sec. 3.2 explains how NOE is manifested in the form of interatomic distance constraints. We will add a brief explanation about the physics of the underlying phenomenon.
> > “inverse problems” feels too broad
>
> We can refine it to “Experiment-guided AlphaFold for characterizing protein structure ensembles” or similar.
> >Need to flip to appendix for results
>
> With more results than can possibly fit into the space limitations, we had to relegate some part to the Appendix. For the text we favored visuals demonstrating conceptually the different test cases. It is not ideal, and we will attempt a better distribution of the results.

---

### Decision · Program_Chairs · 2025-05-01

**Decision:**

Accept (poster)

**Comment:**

The authors propose a framework for building protein conformational ensembles that are consistent with measured experimental data. The authors use Alphafold 3 as a core building block used to generated conformations, and propose an elaborate architecture on top of this to curate the ensemble that relies on diffusion guidance. Reviewers are all supportive of this work, notably post-rebuttal. Some reviewers initially voiced concerns that this work was too close to applications to be relevant to ICML, but these concerns were alleviated following a discussion with authors that emphasized the involvement of the ML community in the study of these problems. I support acceptance.